# Inositol treatment inhibits medulloblastoma through suppression of epigenetic-driven metabolic adaptation

Sara Badodi [1], Nicola Pomella[1], Xinyu Zhang[1], Gabriel Rosser [1], John Whittingham[2], Maria Victoria Niklison-Chirou [1,9], Yau Mun Lim [3], Sebastian Brandner [3], Gillian Morrison[4], Steven M. Pollard [4], Christopher D. Bennett[5,6], Steven C. Clifford[7], Andrew Peet[5,6], M. Albert Basson [2,8] & Silvia Marino [1]✉

Deregulation of chromatin modifiers plays an essential role in the pathogenesis of medulloblastoma, the most common paediatric malignant brain tumour. Here, we identify a BMI1-dependent sensitivity to deregulation of inositol metabolism in a proportion of medulloblastoma. We demonstrate mTOR pathway activation and metabolic adaptation specifically in medulloblastoma of the molecular subgroup G4 characterised by a BMI1High; CHD7Low signature and show this can be counteracted by IP6 treatment. Finally, we demonstrate that IP6 synergises with cisplatin to enhance its cytotoxicity in vitro and extends survival in a pre-clinical BMI1High;CHD7Low xenograft model.

[1] Blizard Institute, Barts and The London School of Medicine and Dentistry, Queen Mary University of London, London, UK. [2] Centre for Craniofacial and Regenerative Biology, King's College London, London, UK. [3] UCL Queen Square Institute of Neurology and The National Hospital for Neurology and Neurosurgery, University College London Hospitals NHS Foundation Trust, London, UK. [4] Centre for Regenerative Medicine & Cancer Research UK Edinburgh Centre, The University of Edinburgh, Edinburgh, UK. [5] Institute of Cancer and Genomic Sciences, University of Birmingham, Birmingham, UK. [6] Birmingham Women and Children's Hospital, Birmingham, UK. [7] Newcastle University Centre for Cancer, Wolfson Childhood Cancer Research Centre, Translational and Clinical Research Institute, Newcastle upon Tyne, UK. [8] MRC Centre for Neurodevelopmental Disorders, King's College London, London, UK. [9] Present address: Centre for Therapeutic Innovation (CTI-Bath), Department of Pharmacy & Pharmacology, University of Bath, Bath, UK. ✉email: s.marino@qmul.ac.uk

Medulloblastoma (MB) is the most common paediatric malignant brain tumour. Large international consortia have assembled substantial cohorts of this relatively rare tumour and have generated comprehensive genomic, epigenomic and transcriptomic datasets, which have informed a novel molecular classification of MB into four distinct molecular subgroups (WNT, SHH, G3 and G4), each of them further subdivided into subtypes with distinct prognosis and responses to therapy[1–4]. Despite these significant advancements in dissecting MB heterogeneity, a translational impact of these observations on how patients are treated, is still lacking. Indeed, the current standard-of-care consists of multimodal therapy, comprising surgery together with radio- and/or chemotherapy[5] that does not account for the specific molecular mechanisms driving tumour growth. This regime cures a substantial proportion of patients, although it is almost invariably associated with severe side effects. Assessment of molecularly tailored therapies, including novel combinations of drugs, which take advantage of the acquired biological knowledge of MB pathogenesis, is now a priority.

Genome-wide sequencing[6–9] has identified mutations in genes regulating essential epigenetic mechanisms, such as histone demethylases, deacetylases, lysine methyltransferases and chromatin remodellers, as well as overexpression of Polycomb-group (PcG) proteins[10–12], particularly in G4 MB, the least understood of all subgroups, despite being the most common and associated with poor prognosis. The PcG protein BMI1, a key regulator of neural stem cell (NSC) self-renewal[13,14], is upregulated in a broad range of cancers, including brain tumours such as glioma[15], where it correlates with advanced clinical stage and poor prognosis[16]. In MB, the highest expression level of BMI1 is associated with the G4 subgroup where it sustains tumour growth[10,17]. We previously demonstrated that high expression of BMI1 together with loss-of-function of the ATP-dependent chromatin remodeller Chromodomain-Helicase-DNA-binding protein 7 (CHD7) cooperates in inducing MB formation when glutamatergic progenitor cells, expressing the proneural transcription factor Math1, are targeted in a sleeping beauty-driven forward genetic screening in the mouse[10]. Moreover, we showed that a BMI1$^{High}$;CHD7$^{Low}$ signature is found in a proportion of G4 MB, a subgroup recently shown to have an ontogenetic link to unipolar brush cell (UBC) progenitor cells, which develop from a Math1$^{+}$ progenitor[18].

Epigenetic mechanisms sustain cancer growth not only through regulation of the expression of genes controlling cell proliferation but also through modulation of pathways involved in cellular metabolism[19]. Energy metabolism is altered in cancer cells, which display enhanced glycolysis favouring production of lactate from glucose even in the presence of oxygen[20] with other metabolic processes, including protein, nucleic acid and lipid biosynthesis, also being enhanced as part of the tumour metabolic reprogramming. Chromatin remodelling via histone modifications and DNA hypermethylation contributes to repression of gluconeogenesis enzymes and consequent activation of glycolysis in different cancers[21–23]. Aerobic glycolysis provides up to 60% of ATP production in MB[24], but whether it is regulated by epigenetic mechanisms in this tumour type is currently unknown.

Energy production is finely tuned to cellular function and state, and is known to be modulated by several signalling pathways, such as Akt/mTOR, in which phosphoinositides and inositol pyrophosphates are key players[25,26]. Inositols (phosphorylated forms of myoinositol and inositol hexakisphosphate, IP6) are highly energetic polyphosphate molecules that play a critical role in a wide variety of physiological and pathological properties of the cell, possibly through phosphorylation-dependent modulation of multiple signalling pathways[27]. Deregulated metabolism of inositol has been described in different diseases, including cancer

where it exerts an antitumour activity mainly modulating cell growth and inducing differentiation and apoptosis[28–30]. However, whether inositol can have therapeutic efficacy and a role in sustaining MB growth has not yet been elucidated.

Here, we perform epigenomic, transcriptomic and proteomic characterisation of patient-derived MB cells modelling the BMI1$^{High}$;CHD7$^{Low}$ molecular convergence, as well as generate genetically engineered and xenograft models to elucidate the molecular mechanisms underpinning BMI1 and CHD7 convergence, its impact on MB metabolism and its druggability at preclinical level.

## Results

**Modelling a BMI1$^{High}$;CHD7$^{Low}$ signature in mice leads to increased number of EOMES$^{+}$ UBC, but no medulloblastoma.** To test the oncogenic potential of the BMI1$^{High}$;CHD7$^{Low}$ molecular signature previously identified[10], we generated mice overexpressing Bmi1 in Math1$^{+}$ progenitors (*Math1Cre;STOP-FloxBmi1*), which were then further crossed with Chd7 floxed mice (*Chd7*$^{flox}$) to obtain *Math1Cre;STOPFloxBmi1;Chd7*$^{f/+}$. Compound mice were healthy and did not develop medulloblastoma during observation of up to 12 months (0/13mice). To assess whether developmental abnormalities were elicited by these genetic modifications, we focussed on neuronal populations originating from Math1$^{+}$ progenitors in the cerebellum. Math1 is expressed in a common progenitor giving rise to both Unipolar Brush Cells (UBC) and Granule Cells (GC)[12,18,31]; therefore we analysed the number of UBC and GC in *Math1Cre;STOP-FloxBmi1;Chd7*$^{f/+}$ compared to single transgenic controls, as assessed by immunostaining for EOMES (Eomesodermin) and NeuN, respectively. NeuN is not expressed in UBC[32]. We observed an increased number of EOMES-expressing UBC in the cerebellum of the compound mutants at both P7 and P21 stages (Fig. 1a, b and Supplementary Fig. S1a–c), whilst the number of mature GC in the cerebellar cortex was reduced (Fig. 1c and Supplementary Fig. S1d). Importantly, no significant changes were observed in the proliferation potential of Granule Cell Progenitors (GCP) at the clonal expansion stage (P7) in these mice (Supplementary Fig. S1e, f).

Next, we interrogated the Human Allen Brain Atlas to map the expression pattern of all genes reported to be expressed, although not necessarily exclusively, in UBC[18] during the development of the human cerebellum. We found increased expression of all genes, including the UBC-specific marker *EOMES*, from the 13th gestational week onwards (Fig. 1d). Expression levels for *BMI1* and *CHD7* were also investigated in a collection of human neural stem cell (hNSC) lines derived from foetal cerebellum at various stages of development (Glioma Cellular Genetics Resource) (Fig. 1e). The hNSC line derived from a 13-week sample had the highest expression of *BMI1* and *CHD7* and this time point was therefore chosen to examine the effect of silencing CHD7 expression, using shRNAs (Supplementary Fig. S1g), on UBC marker expression. We found increased expression of *EOMES* (Fig. 1f) in hNSC upon CHD7 silencing in a BMI1-expressing context, which led to increased proliferation of the cells (Fig. 1g), but no MB formation upon orthotopic injection into the neonatal cerebellum (0/10 mice at 8.5 months).

We show that a BMI1$^{High}$;CHD7$^{Low}$ signature leads to an increased number of UBC and reduced number of mature GC in the developing cerebellum, raising the possibility that it favours the differentiation of a common Math1$^{+}$ progenitor towards a UBC lineage. UBC is the most promising putative cell of origin of G4 MB; hence, our results represent a strong indicator of a potential role for BMI1 and CHD7 in the pathogenesis of this MB subgroup, although they are not sufficient on their own to elicit

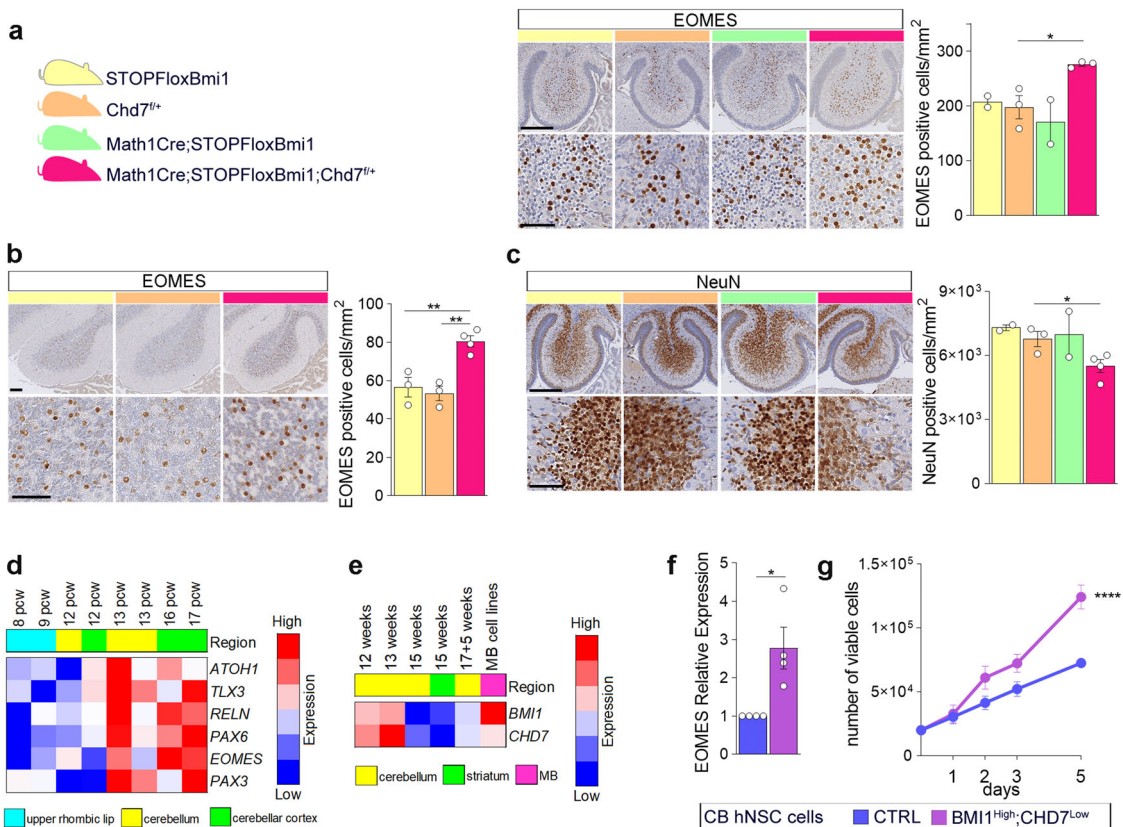

**Fig. 1 Increased number of UBC and expression of EOMES upon modelling of the BMI1^High;CHD7^Low signature in progenitor cells. a, b** Colours identify STOPFloxBmi1 (yellow), Chd7^f/+ (orange), Math1Cre;STOPFloxBmi1 (green) or Math1Cre;STOPFloxBmi1;Chd7^f/+ (pink) genotypes throughout the figure. EOMES IHC staining and quantification of UBC in the cerebellum at P7 (**a**) or P21 (**b**) developmental stages. **a** $n = 2$ biological independent animals per STOPFloxBmi1 and Math1Cre;STOPFloxBmi1 genotypes, $n = 3$ biological independent animals per Chd7^f/+ and Math1Cre;STOPFloxBmi1;Chd7^f/+ genotypes. **b** $n = 3$ biological independent animals per STOPFloxBmi1 and Chd7^f/+ genotypes, $n = 4$ biological independent animals per Math1Cre;STOPFloxBmi1;Chd7^f/+ genotype. One-way ANOVA. **c** NeuN IHC staining and quantification of GC in the cerebellum at P7. $n = 2$ biological independent animals per STOPFloxBmi1 and Math1Cre;STOPFloxBmi1 genotypes, $n = 3$ biological independent animals per Chd7^f/+ genotype, $n = 4$ biological independent animals per Math1Cre; STOPFloxBmi1;Chd7^f/+ genotype, one-way ANOVA. **d** Heatmap showing z-scores of UBC marker expression in the human upper rhombic lip (light blue), cerebellum (yellow) and cerebellar cortex (green) from 8 to 17 pcw. **e** Heatmap showing relative expression of *BMI1* and *CHD7* in a collection of hNCS lines isolated from the cerebellum (yellow) or striatum (green) as compared to MB cell lines (pink). **f** qPCR analysis of *EOMES* expression level in hNSC upon CHD7 silencing (BMI1^High;CHD7^Low, purple) compared to control (CTRL, violet). $n = 4$ biologically independent experiments, two-tailed unpaired t test. **g** Cell proliferation assay of CDH7-silenced or control cells. $n = 6$ biologically independent experiments, two-way ANOVA. All graphs report mean ± SEM. P values: $*P < 0.05$, $**P < 0.01$ or $****P < 0.0001$. Scale bars = 100 μm. Source data are provided as a Source Data file.

MB formation in either genetically engineered mice or hNSC xenografted models.

**mTOR signalling is activated in G4 MB lines, but not cerebellar hNSC, with a BMI1^High;CHD7^Low signature.** To gain further insights into the differences between the role played by the BMI1;CHD7 molecular convergence in a neoplastic and non-neoplastic context, we performed an integrated analysis of the methylome and transcriptome of ICb1299 and CHLA-01-Med MB cell lines. These cell models were chosen because no stable patient-derived xenografts (PDX) have been obtained from BMI1^High;CHD7^Low G4 patients, rendering in vitro genetic modification of BMI1-overexpressing cell lines the only suitable model to assess the contribution of the signature to MB pathogenesis. ICb1299 and CHLA-01-Med MB cell lines display a G3/G4 subgroup affiliation[10,31,33–35], reflecting a widely recognised boundary of plasticity between these two subgroups[31,36]. Modelling of the signature was achieved in these lines by CHD7 silencing in the context of BMI1 overexpression as compared to normal cerebellum as previously described[10]. Integration

of the datasets of the cell model with those obtained in G4 MB tumour samples[4] ensured that the focus would be on events shared between these cell models and subgroup-specific tissue samples (Fig. 2a). We compared samples with and without the signature and identified 214 differentially expressed genes (DEG) with 361 differentially methylated probes (DMP) in MB cells and 2454 DEG with 138 DMP in G4 MB tissues (Supplementary Fig. S2a). Canonical pathway analysis identified significantly enriched pathways related to both signalling and metabolism, including lipids and phospholipid metabolism and energy production, in each of the four datasets (Fig. 2b and Supplementary Fig. S2b). Comparative analysis of the results revealed a single common and conserved pathway Superpathway of Inositol Phosphate Compounds (Fig. 2c), which was not identified when significant DEG between G3 MB tumours with and without BMI1^High;CHD7^Low signature was analysed (Supplementary Fig. S2c), in keeping with the correlation between the signature and this metabolic pathway being specifically pertinent to the G4 subgroup.

Phosphorylated forms of myoinositol and inositol hexakisphosphate (IP6) are highly energetic polyphosphate molecules regulating many physiological cell properties through modulation

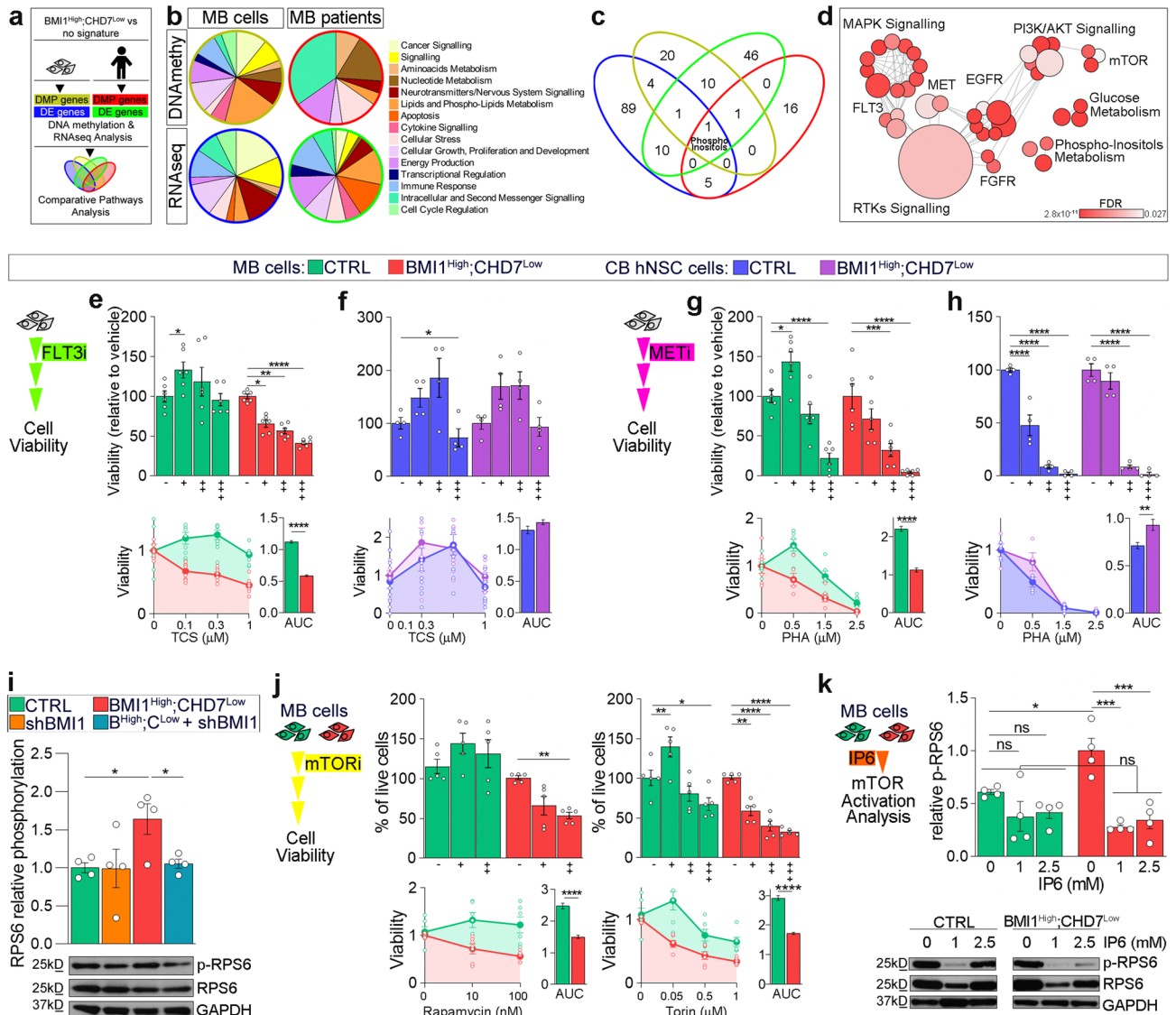

**Fig. 2 Phosphoinositol metabolism is deregulated in BMI1$^{High}$;CHD7$^{Low}$ G4 MB and activation of the mTOR pathway is observed specifically in MB cells modelling this subgroup. a** Schematic representation of the comparative analysis between BMI1$^{High}$;CHD7$^{Low}$ MB cell lines and G4 patients with and without the signature performed to identify genes differentially expressed (DE) and differentially methylated probes (DMP) and commonly deregulated pathways. **b** Pie charts representing canonical pathways differentially enriched between MB cells or patients with or without BMI1$^{High}$;CHD7$^{Low}$ signature identified in RNA-Seq or DNA methylation analysis. Canonical pathways are classified based on IPA categories list and colour-coded accordingly. **c** Venn diagram of canonical pathways enriched for genes differentially expressed (blue and light-green diagrams as coloured in **a** and **b**) or methylated (olive green and red as coloured in **a** and **b**) in BMI1$^{High}$;CHD7$^{Low}$ cell lines and GR4 MBs with the matching signature. Highlighted the superpathway of phosphoinositol compounds (phosphoinositols), which is common to all the datasets analysed. **d** Bubble plot showing significant Reactome pathways obtained in phosphoproteomic analysis of BMI1$^{High}$;CHD7$^{Low}$ MB lines. Bubbles are coloured based on FDR values, and size is proportional to the number of genes of specific pathways. **e**, **f** Cell viability assays of MB cells (**e**) or hNSC (**f**) with (BMI1$^{High}$;CHD7$^{Low}$) or without the signature (CTRL) upon 72 h of treatment with increasing concentrations of FLT3 inhibitor (TCS). Histograms represent the percentages of viable cells relative to non-treated cells (top). Measurement of Area Under Curve (AUC) (bottom) to compare the overall response to treatment. **e** n = 6 biologically independent experiments, (**f**) n = 4 biologically independent experiments, two-way ANOVA. **g**, **h** Cell viability assays of MB cells (**g**) or hNSC (**h**) with (BMI1$^{High}$;CHD7$^{Low}$) or without the signature (CTRL) upon 72 h of treatment with increasing concentrations of MET inhibitor (PHA). Histograms represent the percentages of viable cells relative to non-treated cells (top). Measurement of the area under the curve (AUC) (bottom) to compare the overall response to treatment. **g** n = 6 biologically independent experiments, **h** n = 4 biologically independent experiments, two-way ANOVA. **i** Western blot and quantification of phosphorylated/total RPS6 (pRPS6, Ser240/244) in MB cells. GAPDH immunoreactivity was used to normalise protein loading. n = 4 biologically independent experiments, one-way ANOVA. **j** Cell viability assays of MB cells with (BMI1$^{High}$;CHD7$^{Low}$, red) or without the signature (CTRL, green) upon 72 h of treatment with increasing concentrations of mTOR pathway inhibitors (rapamycin and torin). Histograms represent the percentages of viable cells relative to non-treated cells (top). Measurement of Area Under Curve (AUC) (bottom) to compare the overall response to treatment. n = 5 biologically independent experiments, two-way ANOVA. **k** Western blot and quantification of phosphorylated/total RPS6 (pRPS6, Ser240/244) after 24 h of treatment with increasing concentrations of IP6 in control (green) or BMI1$^{High}$;CHD7$^{Low}$ (red) MB cells. GAPDH immunoreactivity was used to normalise protein loading. n = 4 biologically independent experiments, two-way ANOVA. All graphs report mean ± SEM. P values: *P < 0.05, **P < 0.01, ***P < 0.001 or ****P < 0.0001. Source data are provided as a Source Data file.

of phosphorylation-dependent pathways[27]. To investigate phosphorylation-dependent events, possibly related to the inositol metabolism, we performed a phosphoproteomic analysis of BMI1$^{High}$;CHD7$^{Low}$ MB lines and controls. We identified 1499 significantly deregulated phosphopeptides, 684 hyper- and 815 hypophosphorylated (Supplementary Fig. S2d), and interrogation of this dataset with Reactome pathway analysis confirmed modulation of phosphoinositol metabolism (Fig. 2d). Moreover, we found an enrichment in signalling mediated by receptor tyrosine-kinase (RTK) and several related pathways, including MAPK/ERK, as previously described[10]. Because this pathway is deregulated also in hNSC[10], we reasoned that it must not be sufficient to trigger MB formation. MET and FLT3 RTKs are also enriched for in our phosphoproteomic analysis (Fig. 2d) with the expression of both receptors being significantly higher in BMI1$^{High}$;CHD7$^{Low}$ MB than in controls, whilst no significant difference was observed in hNSC (Supplementary Fig. S2e, f). To assess the functional impact of the observed MB-specific increased expression of these two RTKs, we analysed cell viability of MB or hNSC cells upon treatment with TCS359 and PHA665752, specific inhibitors of FLT3 and MET, respectively. Pharmacologic inhibition of FLT3 impaired cell viability specifically in MB modelling the signature (Fig. 2e), without affecting hNSC (Fig. 2f). Moreover, BMI1$^{High}$;CHD7$^{Low}$ MB cells were more susceptible to MET inhibition (Fig. 2g), although hNSC was also affected, albeit in the opposite direction, i.e. those modelling the signature were more resistant (Fig. 2h). These results validate the in silico prediction of enhanced signalling through the tyrosine-kinase receptors FLT3 and MET in BMI1$^{High}$;CHD7$^{Low}$ MB but not in hNSC with the same molecular signature.

RTKs modulate a variety of downstream phosphorylation-dependent signalling cascades[37,38]. Integrated pathway analysis of DE genes and phosphorylation-dependent proteins predicted an impact on mTOR signalling in BMI1$^{High}$;CHD7$^{Low}$ MB, a modulation that is also observed in tumour samples with the same signature (Fig. 2d and Supplementary Fig. S2h). To validate this in silico prediction, we analysed phosphorylation of ribosomal protein S6 (RPS6), a well-characterised target of the mTOR pathway. We confirmed the overactivation of the mTOR pathway in MB cells modelling the signature, which is dependent on both CHD7 silencing and concomitant expression of BMI1 (Fig. 2i and Supplementary Fig. S2f, g). FLT3 and MET activate AKT/mTOR signalling[39–41]. In line with the observation that BMI1$^{High}$;CHD7$^{Low}$ MB cells show an increased dependence on FLT3 and MET signalling, we found that their inhibition decreases mTOR activation specifically in BMI1$^{High}$;CHD7$^{Low}$ MB cells (Supplementary Fig. S2i, j). Importantly, no impact on RPS6 phosphorylation was observed in BMI1$^{High}$;CHD7$^{Low}$ hNSC (Supplementary Fig. S2k), raising the possibility that the impact on the mTOR pathway is MB-specific. Impaired cell viability was observed upon treatment with two mTOR inhibitors (rapamycin and torin) in BMI1$^{High}$;CHD7$^{Low}$ MB as compared to control (Fig. 2j and Supplementary Fig. S2i) and IP6 treatment reduced mTOR activation specifically in BMI1$^{High}$;CHD7$^{Low}$ MB cells as compared to cells without the signature (Fig. 2k).

Our data show deregulation of inositol metabolism and RTKs in MB cells and G4 tumour samples with a BMI1$^{High}$;CHD7$^{Low}$ signature and activation of the mTOR signalling pathway in these cells, but not in hNSC with the same molecular signature.

## A BMI1$^{High}$;CHD7$^{Low}$ signature induces metabolic adaptation in MB but not in hNSC

Both phosphoinositols and mTOR pathways regulate cell viability primarily via modulation of energy production[42–45]. We, therefore, studied the energetic state

of BMI1$^{High}$;CHD7$^{Low}$ MB cells and analysed mitochondrial respiration and glycolysis, by measuring oxygen consumption rate (OCR) and extracellular acidification rate (ECAR), respectively. OCR was decreased in BMI1$^{High}$;CHD7$^{Low}$ MB cells with a reduction in basal, ATP-linked and maximal respiration, indicating an impaired mitochondrial function (Fig. 3a and Supplementary Fig. S3a) rather than a reduced number of mitochondria, as no significant changes in mtDNA were detected (Supplementary Fig. S3b). Consistently, staining with a potential-dependent MitoTracker dye showed increased mitochondrial membrane potential, suggestive of mitochondrial hyperpolarisation in BMI1$^{High}$;CHD7$^{Low}$ MB cells (Supplementary Fig. S3c). Intriguingly, mitochondrial dysfunction is among the significantly enriched pathways found in BMI1$^{High}$;CHD7$^{Low}$ G4 MB tissue samples (Supplementary Fig. S2b). In contrast, increased ECAR with enhanced glycolysis and glycolytic capacity was detected in MB cells with the signature (Fig. 3b). These results indicate metabolic adaptation in BMI1$^{High}$;CHD7$^{Low}$ MB cells, leading to impaired mitochondrial respiration and enhanced aerobic glycolysis.

To assess whether this finding could be useful for future use in the clinic, we identified BMI1$^{High}$;CHD7$^{Low}$ G4 MB in a cohort of patients where magnetic resonance spectroscopy (MRS) analysis had been performed on surgically resected tumour samples[46]. We show an increased quantity of leucine and valine, two branched-chain amino acids (BCAA), in MB with the signature as compared to samples without the signature, in keeping with an enhanced glycolytic status also in the tumour tissue (Supplementary Fig. S3d). Consistently, we found increased expression of *HK2*, *PFKP*, *ENO4*, *PDK1* and *LDHB*, encoding key enzymes regulating glucose metabolism, in BMI1$^{High}$;CHD7$^{Low}$ G4 MB tumour samples (Supplementary Fig. S3e) but not in G3 tumours with the same signature (Supplementary Fig. S3f), further confirming the relevance of the BMI1$^{High}$;CHD7$^{Low}$ signature specifically for G4 MB subgroup.

Mitochondria and glycolysis produce ATP; hence, we determined the total quantity of ATP produced in a steady state and found that BMI1$^{High}$;CHD7$^{Low}$ MB cells produce significantly more ATP than controls (Fig. 3c). Moreover, the analysis of ADP/ATP ratio showed no significant difference (Fig. 3d), indicating that ATP utilisation is not different between MB cells with or without the signature and suggesting that the observed deregulation of OCR and ECAR ultimately impacted energy production. Importantly, IP6 treatment reverted the metabolic adaptation observed in BMI1$^{High}$;CHD7$^{Low}$ MB cells by decreasing ECAR and glycolytic capacity (Fig. 3e), without affecting OCR (Supplementary Fig. S3g). In contrast, BMI1$^{High}$;CHD7$^{Low}$ hNSC showed increased mitochondrial (Fig. 3f and Supplementary Fig. S3h, i) and glycolytic functions (Fig. 3g) compared to isogenic hNSC without the signature. However, BMI1$^{High}$;CHD7$^{Low}$ hNSC show a reduced total ATP production (Fig. 3h) possibly due to increased ATP utilisation as suggested by increased ADP/ATP ratio compared to hNSC without the signature (Fig. 3i). Interestingly, IP6 treatment does not affect ECAR, glycolysis and glycolytic capacity of BMI1$^{High}$;CHD7$^{Low}$ hNSC (Supplementary Fig. S3j), indicating that IP6-mediated inhibition of glycolytic function is achieved specifically in MB context.

Taken together, our data show that a BMI1$^{High}$;CHD7$^{Low}$ signature induces a metabolic adaptation in G4 MB, but not in hNSC with the same signature, and that it can be counteracted by IP6 treatment.

## IP6 sensitivity in BMI1$^{High}$;CHD7$^{Low}$ MB cells is BMI1-dependent

Taking into account that inositol pathway dysregulation, through IP6 treatment, inhibits mTOR pathway activation and

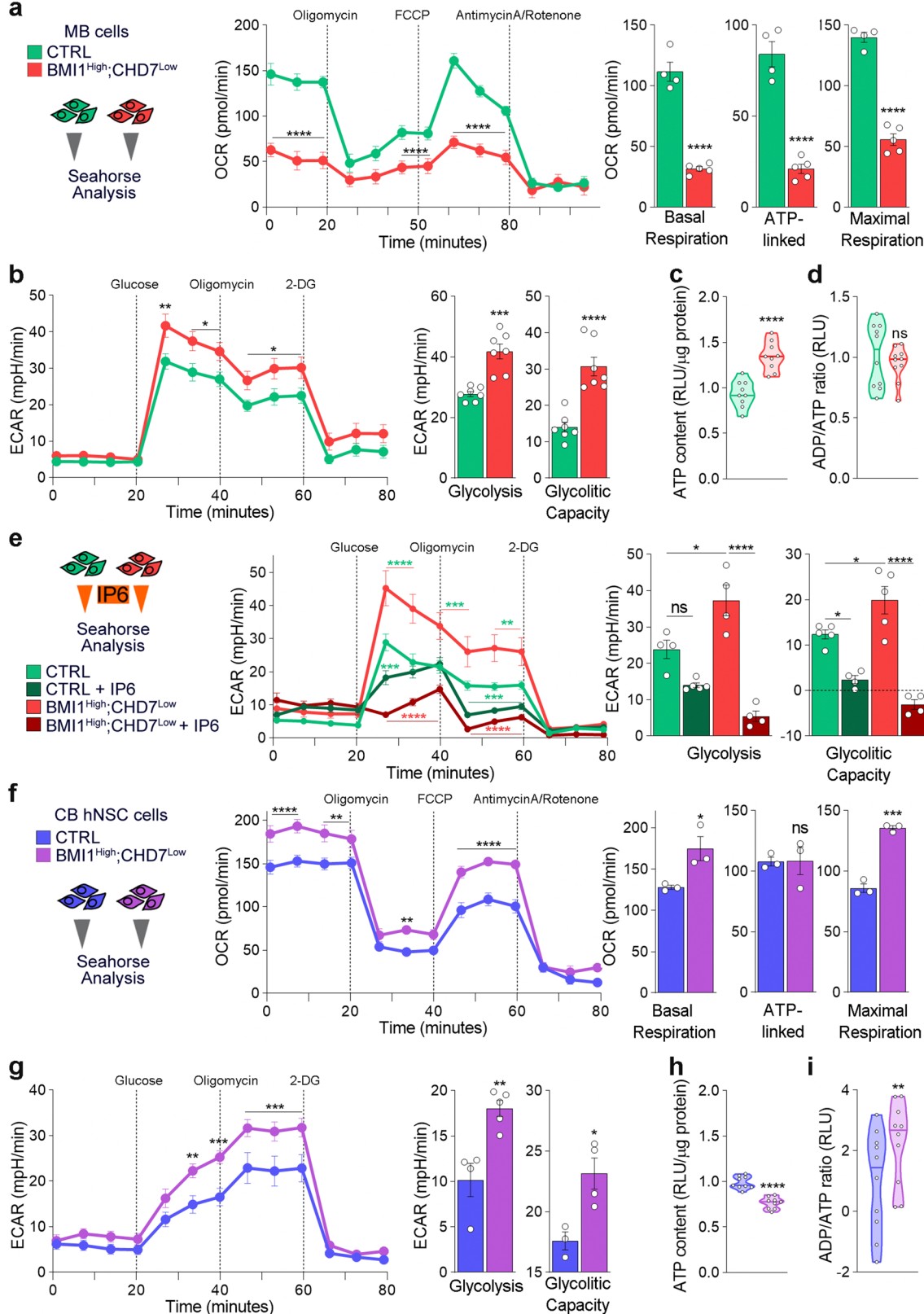

the metabolic adaptation mediated by the signature in G4 MB cells, we set out to assess their impact on tumour cell viability. We found impaired cell proliferation in BMI1$^{High}$;CHD7$^{Low}$ MB lines at all tested IP6 concentrations compared to control MB cells without the signature, which were only affected by the highest concentration used (Fig. 4a, b and Supplementary Fig. S4a). No induction of

apoptosis was found upon treatment (Supplementary Fig. S4b), suggesting a cytostatic rather than a cytotoxic effect mediated by IP6. In contrast, treatment of hNSC with or without the signature did not affect cell viability (Fig. 4c), raising the possibility that IP6 could be a novel signature-specific treatment for MB with a very low toxicity on the surrounding developing cerebellum.

**Fig. 3 Metabolic adaptation in BMI1$^{High}$;CHD7$^{Low}$ MB cells can be reverted by inositol treatment. a** Analysis of oxidative consumption rate (OCR) using seahorse extracellular flux assay. BMI1$^{High}$;CHD7$^{Low}$ (red) or control (green) MB cells were sequentially treated with oligomycin, P-trifluoromethoxy carbonyl cyanide phenylhydrazone (FCCP) or a combination of antimycin A and rotenone at the indicated time points (dashed lines). Histograms (right) show OCR production during basal, ATP-linked and maximal respiration. $n = 4$ biologically independent experiments for CTRL or $n = 5$ biologically independent experiments for BMI1$^{High}$;CHD7$^{Low}$, two-tailed unpaired $t$ test. **b** Analysis of extracellular acidification rate (ECAR) using seahorse extracellular flux assay. BMI1$^{High}$;CHD7$^{Low}$ or control MB cells were sequentially treated with glucose, oligomycin or 2-deoxy-D-glucose (2DG) at the indicated time points (dashed lines). Histograms (right) show ECAR representing glycolysis and glycolytic capacity. $n = 7$ biologically independent experiments, two-tailed unpaired $t$ test. **c** Quantification of ATP content in BMI1$^{High}$;CHD7$^{Low}$ or control MB cells, normalised on quantity of total protein. $n = 9$ biologically independent experiments, two-tailed unpaired $t$ test. **d** Quantification of ADP/ATP ratio in BMI1$^{High}$;CHD7$^{Low}$ or control MB cells. $n = 10$, two-tailed unpaired $t$ test. **e** Analysis of ECAR using seahorse extracellular flux assay. BMI1$^{High}$;CHD7$^{Low}$ or control MB cells were incubated for 24 h with IP6 1 mM and then sequentially treated with glucose, oligomycin or 2-deoxy-D-glucose (2DG) at the indicated time points (dashed lines). Histograms (right) show ECAR representing glycolysis and glycolytic capacity. $n = 4$ biologically independent experiments, one-way ANOVA. **f** Analysis of oxidative consumption rate (OCR) using seahorse extracellular flux assay. BMI1$^{High}$;CHD7$^{Low}$ (purple) or control (violet) hNSC cells were sequentially treated with oligomycin, FCCP or a combination of antimycin A and rotenone at the indicated time points (dashed lines). Histograms (right) show OCR production during basal, ATP-linked and maximal respiration. $n = 3$ biologically independent experiments, two-tailed unpaired $t$ test. **g** Analysis of extracellular acidification rate (ECAR) using seahorse extracellular flux assay. BMI1$^{High}$;CHD7$^{Low}$ or control hNSC cells were sequentially treated with glucose, oligomycin or 2-deoxy-D-glucose (2DG) at the indicated time points (dashed lines). Histograms (right) show ECAR representing glycolysis and glycolytic capacity. $n = 3$ biologically independent experiments, two-tailed unpaired $t$ test. **h** Quantification of ATP content in BMI1$^{High}$;CHD7$^{Low}$ or control hNSC cells, normalised on the quantity of total protein. $n = 9$ biologically independent experiments, two-tailed unpaired $t$ test. **i** Quantification of ADP/ATP ratio in BMI1$^{High}$;CHD7$^{Low}$ or control hNSC cells. $n = 10$ biologically independent experiments, two-tailed unpaired $t$ test. All graphs report mean ± SEM. $P$ values: *$P < 0.05$, **$P < 0.01$, ***$P < 0.001$ or ****$P < 0.0001$. Source data are provided as a Source Data file.

## A PRC (Polycomb Repressive Complex)-dependent CHD7-mediated epigenetic regulation of lipid inositol signalling in BMI1$^{High}$;CHD7$^{Low}$ MB cells.

Because BMI1 expression mediated IP6 sensitivity in a context of low CHD7 expression (Fig. 4a, b and Supplementary Fig. S4a), we set out to clarify the signature-specific role of BMI1 in the inositol- related response. We interrogated the genome-wide distribution of BMI1 in the two MB cell lines using chromatin immunoprecipitation followed by deep sequencing (ChIP-Seq) and focussed on the shared peaks. We found a significant redistribution of BMI1 binding across the genome in a CHD7$^{Low}$ cellular context, with a two-fold enrichment in the percentage of promoter occupancy (Fig. 4d). Next, we assessed differential BMI1 binding at the promoter of genes related to phosphoinositol metabolism upon CHD7 silencing and identified FLT3, MET and phosphatidylinositol-4-phosphate 5-kinase type 1 beta (PIP5K1B) uniquely bound by BMI1 in control cells with the expression of FLT3 and MET increasing upon BMI1 silencing (Supplementary Fig. S4c), suggesting a direct regulation via BMI1. To link the observation of BMI1 binding to a PRC (Polycomb Repressive Complex)-mediated epigenetic regulation, we identified those promoters that were also marked by histone H3 lysine 27 trimethylation (H3K27me3) and found a significant overlap in MB cells with and without signature (210 and 475 representing 76.1 and 66.4% of all BMI1 peaks on promoters, respectively, Fisher's exact test: $p < 0.0001$) (Fig. 4e and Supplementary Fig. S4d). Gene Ontology (GO) enrichment analysis of the genes with BMI1 and H3K27me3 promoter co-occupancy identified activation of phospholipase C (PLC) activity specifically in cells with the signature (Fig. 4e and Supplementary Fig. S4e). PLC produces diacylglycerol (DAG) and inositol 1,4,5-trisphosphate (IP3) and consequently regulates inositol lipid signalling pathways[47,48]. Thus, our data suggest a BMI1-dependent CHD7-mediated impact on inositol signalling in MB cells.

## A BMI1-mediated dynamic epigenetic regulation of inositol lipid signalling through poised promoters in BMI1$^{High}$;CHD7$^{Low}$ MB cells.

Because chromatin remodelling factors can dynamically regulate gene expression through bivalent domains on promoters marked by both the repressive H3K27me3 and the activating trimethylation on lysine 4 of histone H3 (H3K4me3) marks[49], we sought to identify genes with BMI1-bound promoter regions also harbouring H3K27me3 and H3K4me3 marks and found 149 and 64 genes representing 20.8% or 23.2% of all BMI1-bound promoters in MB cells without or with BMI1$^{High}$;CHD7$^{Low}$ signature. As expected, GO enrichment analysis revealed a shared enrichment for biological processes related to nervous system development and neuron fate commitment (Fig. 4f and Supplementary Fig. S4f), in line with the observations of bivalent domains mainly marking key developmental genes[49]. Intriguingly, we also found activation of PLC activity enriched for in BMI1$^{High}$;CHD7$^{Low}$ MB (Fig. 4f and Supplementary Fig. S4f), suggesting a BMI1-mediated dynamic epigenetic regulation of inositol lipid signalling through poised promoters. Finally, we assessed whether any of the 64 genes with the promoter co-occupied by BMI1, H3K27me3 and H3K4me3 were related to the Superpathway of Phosphoinositol Compounds. PPIP5K2, an inositol pyrophosphate kinase that synthesises 1-IP7 from IP6[50], was identified specifically in BMI1$^{High}$;CHD7$^{Low}$ MB cells, with the H3K27me3 mark being lost at its promoter upon BMI1 silencing (Fig. 4g and Supplementary Fig. S4g). Because PPIP5K2 expression was downregulated upon IP6 administration specifically in BMI1$^{High}$;CHD7$^{Low}$ MB cells (Fig. 4g), our data raise the possibility that epigenetic regulation of PPIP5K2 regulates inositol metabolism and enhances IP6-induced cytotoxicity in these cells. Moreover, inositol-related canonical pathways enriched for genes differentially methylated upon CHD7 silencing were found to switch from a hyper- to hypomethylated status (Fig. 4h) upon concomitant BMI1 silencing, including PPIP5K2 (Supplementary Fig. S4h), providing additional support to the conclusion that the observed BMI1-dependent deregulation of inositol metabolism is epigenetically regulated (Supplementary Fig. S4i).

In conclusion, our data show that the BMI1$^{High}$;CHD7$^{Low}$ specific response to inositol pathway alteration, through IP6 treatment, is dependent on BMI1-mediated epigenetic regulation.

## IP6 synergises with cisplatin to reduce cell viability in BMI1$^{High}$;CHD7$^{Low}$ MB cells.

Cisplatin and etoposide are chemotherapeutic agents currently used in the treatment of MB patients[5]. To begin assessing the potential value of inositol treatment as adjuvant therapy, we treated BMI1$^{High}$;CHD7$^{Low}$ MB cells with either of the two chemotherapeutic agents alone or in combination with IP6. We found that control cells are resistant

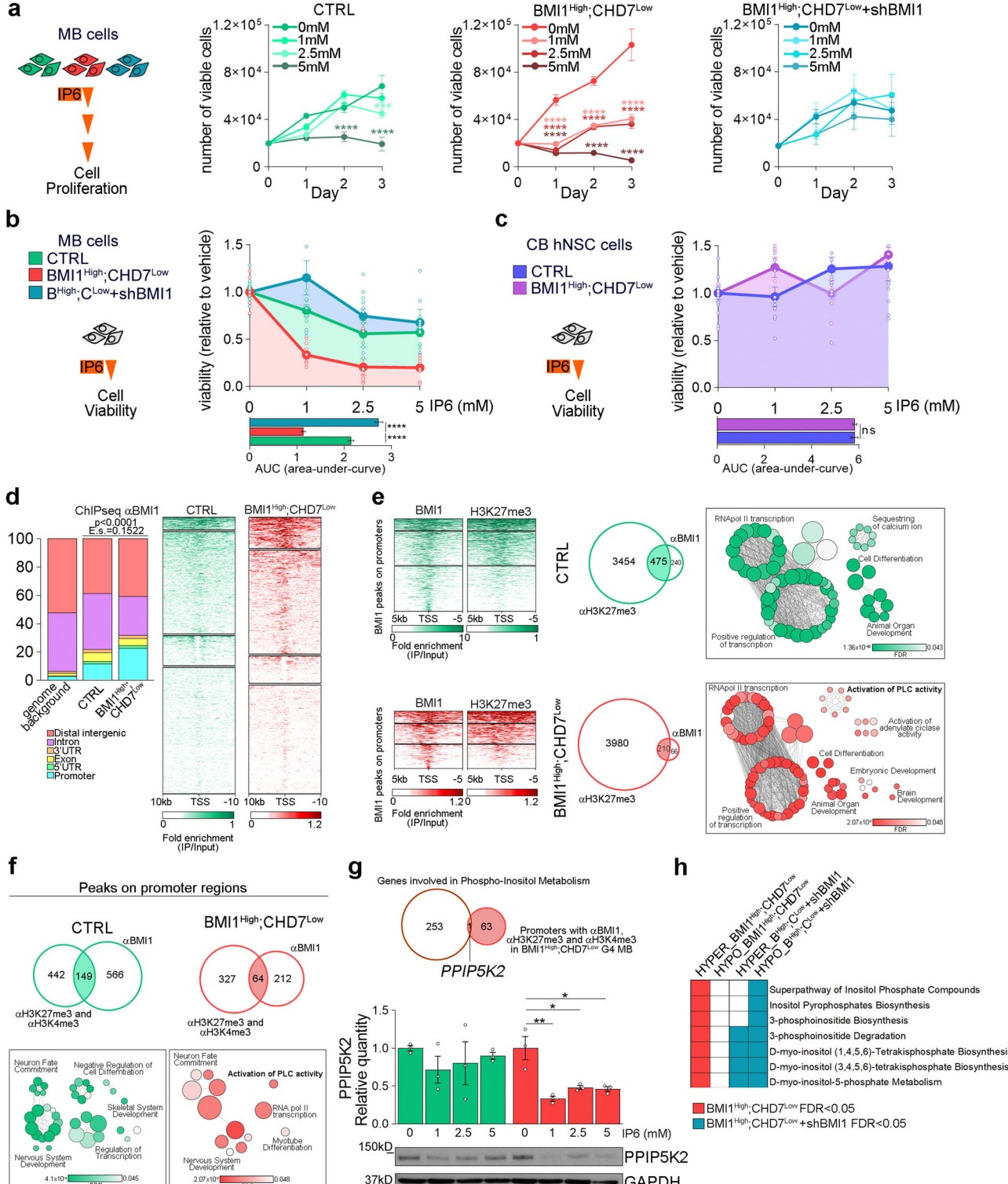

to the range of cisplatin concentration used in vitro, while treatment with etoposide caused a decreased cell viability (Fig. 5a). In contrast, BMI1$^{High}$;CHD7$^{Low}$ cells are sensitive to all the tested doses of cisplatin, while etoposide only induces decreased cell viability at the highest concentration (Fig. 5a). Combination treatments with IP6 showed that it synergises with cisplatin, but not etoposide, enhancing its cytotoxic activity specifically in BMI1$^{High}$;CHD7$^{Low}$ cells (Fig. 5b, c).

Altogether, these data provide evidence of a synergic effect of IP6 and cisplatin in BMI1$^{High}$;CHD7$^{Low}$ MB in vitro.

**Combined IP6 and cisplatin treatment improves survival of BMI1$^{High}$;CHD7$^{Low}$ MB xenografts.** Leveraging the evidence that inositol administration through drinking water can rescue a brain-related lethal knockout phenotype[51] and is therefore reaching the brain even when the blood–brain barrier (BBB) is intact, we set out to assess whether IP6 treatment could synergise with cisplatin also in a preclinical in vivo model of MB.

BMI1$^{High}$;CHD7$^{Low}$ and control MB cells were injected into the cerebellum of newborn mice, which were treated with IP6 either alone or in combination with cisplatin at three weeks of

**Fig. 4 IP6 sensitivity in BMI1$^{High}$;CHD7$^{Low}$ MB cells is BMI1-dependent. a** Cell proliferation assays of control (CTRL, green), BMI1$^{High}$;CHD7$^{Low}$ (red) or with concomitant BMI1 silencing (BMI1$^{High}$;CHD7$^{Low}$ + shBMI1, blue) MB cells upon 72 h and treated with increasing concentrations of IP6. $n = 6$ biologically independent experiments, two-way ANOVA used to calculate p-values (colour-coded) relative to not treated cells. **b** Cell viability assays of control, BMI1$^{High}$;CHD7$^{Low}$ or with concomitant BMI1 silencing MB cells upon 24 h of treatment with increasing concentrations of IP6. Measurement of the area under the curve (AUC) to compare the overall response to treatment. $n = 6$ biologically independent experiments, one-way ANOVA. **c** Cell viability assays of control (violet) or BMI1$^{High}$;CHD7$^{Low}$ (purple) hNSC upon 24 h of treatment with increasing concentrations of IP6. Measurement of the area under the curve (AUC) to compare the overall response to treatment. $n = 6$ biologically independent experiments, two-tailed unpaired t test. **d** Distribution of genomic annotations and heatmaps of BMI1 ChIP-Seq peak enrichments in control (CTRL) or BMI1$^{High}$;CHD7$^{Low}$ MB cells. Heatmaps are centred at TSS and ordered by BMI1 intensity. Signals were clustered based on the distribution of BMI1 surrounding the promoters. One-sided $\chi^2$ test $P$ value and effect size (E.s.) are reported. **e** Heatmaps of BMI1 and H3K27me3 ChIP-Seq peak enrichment on BMI1-bound promoters in CTRL or BMI1$^{High}$;CHD7$^{Low}$ MB cells (left panels). Heatmaps are centred at TSS and ordered by BMI1 intensity. Signals were clustered based on the distribution of BMI1 and H3K27me3 surrounding the promoters. Venn diagrams showing the overlap between genes with BMI1 and H3K27me3 on their promoter (middle panels). Bubble plots showing GO biological processes significantly enriched for genes with promoter co-bound by BMI1 and H3K27me3 in control (green) or BMI1$^{High}$;CHD7$^{Low}$ (red) MB cells (right panels). Bubbles are coloured based on FDR values and size is proportional to the number of genes of specific GO terms. **f** Venn diagrams showing the overlap between genes with BMI1, H3K27me3 and H3K4me3 on their promoter (top panels). Bubble plots showing GO biological processes significantly enriched for genes with promoter co-bound by BMI1, H3K27me3 and H3K4me3 in control (green) or BMI1$^{High}$; CHD7$^{Low}$ (red) G4 MB cells (bottom panels). Bubbles are coloured based on FDR values and size is proportional to the number of genes of specific GO terms. **g** Venn diagram showing the overlap between a gene involved in phosphoinositol compound metabolism and gene with BMI1, H3K27me3 and H3K4me3 on their promoter (top) in BMI1$^{High}$;CHD7$^{Low}$ MB cells. Western blot and quantification of PPIP5K2 expression after 24 h of treatment with increasing concentrations of IP6 in control (green) or BMI1$^{High}$;CHD7$^{Low}$ (red) MB cells. GAPDH immunoreactivity was used to normalise protein loading. $n = 3$ biologically independent experiments, two-way ANOVA. **h** Heatmap showing inositol-related pathways significantly enriched for genes differentially methylated in BMI1$^{High}$;CHD7$^{Low}$ (red) or BMI1$^{High}$;CHD7$^{Low}$ with concomitant BMI1 silencing (blue). All graphs report mean ± SEM. $P$ values: $*P < 0.05$, $**P < 0.01$ or $****P < 0.0001$. Source data are provided as a Source Data file.

age, with cisplatin being administered over a three-week period and IP6 until symptoms occurred (Fig. 5d). We observed significantly shorter survival in mice xenografted with BMI1-High;CHD7$^{Low}$ MB as compared to control mice (Supplementary Fig. S5a), in keeping with our previous observation that patients with BMI1$^{High}$;CHD7$^{Low}$ G4 MB have a poorer prognosis than other G4 MB patients[10]. Assessment of tumour size confirmed that these tumours were larger than control xenografts (Fig. 5g). Next, we examined the effect of the treatment on the survival of the mice xenografted with BMI1$^{High}$;CHD7$^{Low}$ MB cells and found that both IP6 and cisplatin were significantly extending survival when compared to untreated mice, in particular the combination therapy led to longer survival as compared to either of the two single agents (IP6 or cisplatin alone) (Fig. 5e and Supplementary Fig. S5b, c). In contrast, xenografts derived from control cells benefited only from cisplatin treatment, with the combination treatment not being significantly different (Fig. 5f and Supplementary Fig. S5b, c). Analysis of the improvement in median survival, achieved in comparison to vehicle treatment, showed doubling of the survival of the BMI1$^{High}$;CHD7$^{Low}$ xenografts compared to controls (without the signature) receiving the same treatment (Supplementary Fig. S5b–e). At a mechanistic level, impaired cell proliferation but no impact on apoptosis was found specifically in tumours originated from MB cells modelling the signature, as assessed by Ki-67 and cleaved Caspase-3 immunostaining (Fig. 5h, i), a finding that recapitulates the in vitro observation of impaired proliferation mediated by IP6 (Fig. 4a and Supplementary Fig. S4b).

In conclusion, these data confirm and extend our in vitro finding that IP6 enhances cisplatin cytotoxic activity only in MB cells modelling the BMI1$^{High}$;CHD7$^{Low}$ signature and provide preclinical evidence that combinatorial treatment could be effective in patients with BMI1$^{High}$;CHD7$^{Low}$ G4 MB.

## Discussion

We have identified a novel epigenetic regulation of inositol metabolism in G4 MB with a BMI1$^{High}$;CHD7$^{Low}$ molecular signature. We show overactivation of the mTOR pathway and induction of metabolic adaptation in G4 MB cells, but not in

hNSC with the same signature, which can be counteracted by IP6 treatment. We provide proof of principle that IP6 synergises with cisplatin enhancing its cytotoxic activity in in vitro and in an in vivo pre-clinical model of BMI1$^{High}$;CHD7$^{Low}$ MB where it significantly extended survival of the xenografted mice.

In a forward genetic screening approach in the mouse, we have previously described a molecular convergence of the chromatin remodellers BMI1 and CHD7, which also characterises a proportion of G4 MB with a significantly decreased overall survival[10]. We have shown overactivation of MAPK/ERK signalling as a downstream effector of this convergence mediating proliferation in MB cell lines and tumour growth in xenografts. Because increased proliferation and overactivation of the MAPK/ERK pathway were observed also in hNSC upon CHD7 silencing in a BMI1-overexpressing context[10], it was conceivable that this was the mechanism underpinning neoplastic transformation of progenitors into a subset of G4 MB. Yet we show here that medulloblastoma did not develop upon modelling the signature in Math1$^+$ progenitor cells in genetically engineered mice. Math1$^+$ progenitor cells were chosen to test this hypothesis because originally transformed in the Sleeping Beauty-driven transposon screen[10] and also because they have been proposed as the cell of origin of a proportion of G4 MB in single-cell transcriptomic studies comparing MB subgroup and progenitor cells during cerebellar development[18,31]. In particular, UBC progenitors which are derived from the Math1$^+$ lineage are remarkably similar at the transcriptomic level to a proportion of G4 MB[18,31]. EOMES, a glutamatergic lineage-specific transcription factor previously described as a possible master regulator of G4 MB subgroup[12], is expressed in UBC cells[18,31]. We show an increased number of EOMES$^+$ UBC and reduced number of NeuN$^+$ granule cells in *Math1Cre;STOPFloxBmi1;Chd7$^{f/+}$* mice suggesting that this signature, albeit not sufficient to elicit MB formation, favours a differentiation toward the UBC lineage rather than the granule cell lineage, potentially predisposing these cells to the acquisition of additional genetic/epigenetic lesions leading to malignant transformation.

BMI1 and CHD7 are chromatin remodelling factors known to modulate gene expression through epigenetic mechanisms[52,53]. Besides their traditional roles as members of the PRC1 complex[54]

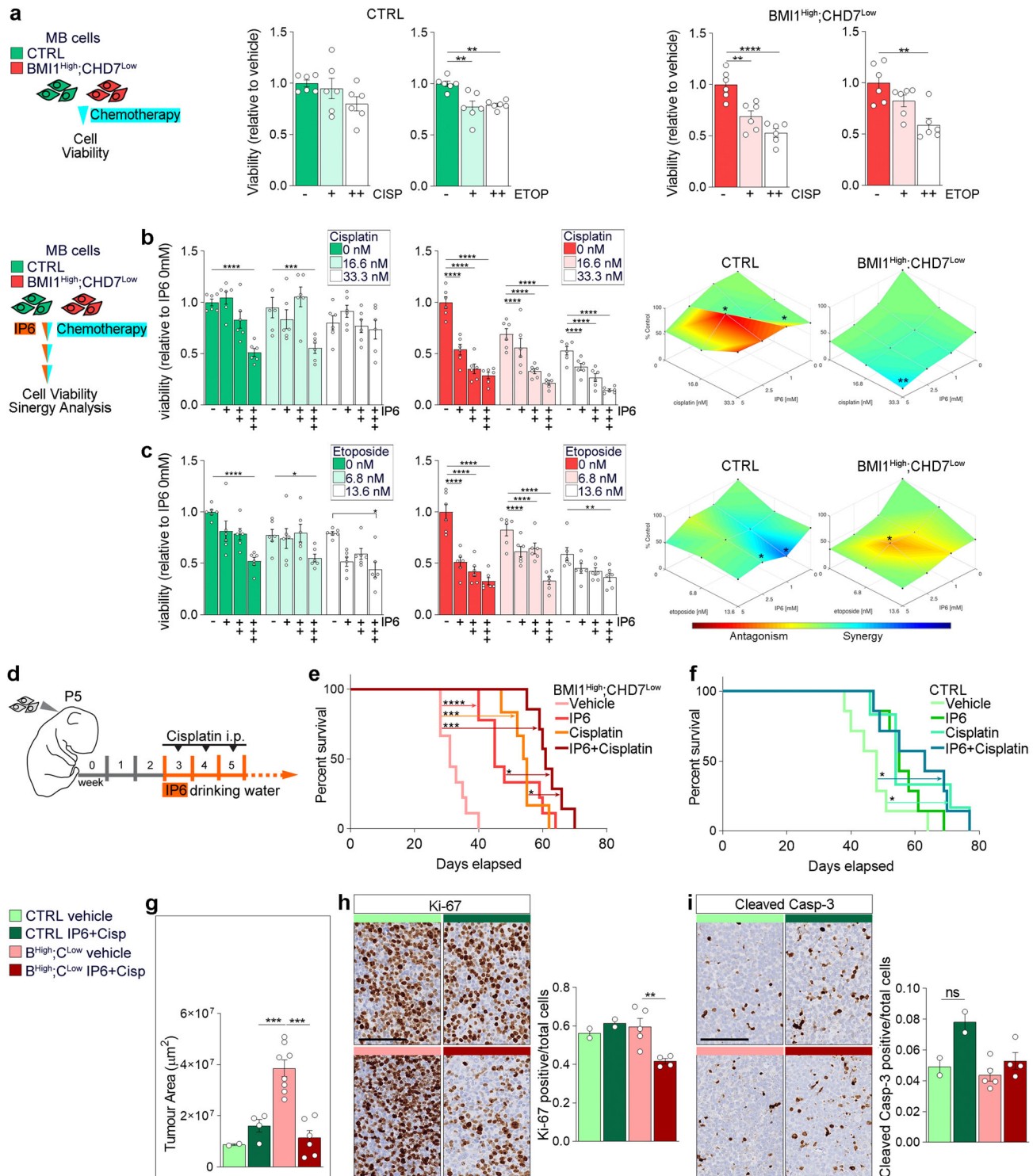

and in the organisation of chromatin structure[55], respectively, several reports have highlighted a direct role of PRC complexes in DNA methylation[56–59] and CHD7 loss-of-function mutations have been associated with specific patterns of CpG DNA methylation modifications[60]. Therefore, to obtain a comprehensive understanding of the effects elicited by a BMI1High;CHD7Low signature in a neoplastic and non-neoplastic cell context, while also taking advantage of the extensive collection of publicly available omics datasets on bulk tumour tissue for comparative analysis, we studied the transcriptome and methylome of MB cells modelling the signature and integrated the results with those

of the G4 bulk tumours. We show a striking convergence on phosphoinositol compounds metabolism when all datasets are integrated, indicating a mechanism shared between the cell model and the tumour bulk with a BMI1High;CHD7Low signature, which is consistent across in vitro and in vivo MB environmental conditions.

The identification of key mechanisms regulated by phosphorylation in BMI1High;CHD7Low G4 MB, together with the recent identification of a G4-specific phosphorylation profile[61,62], prompted us to widen our analysis by analysing all phospho-regulated proteins in BMI1High;CHD7Low MB cells by mass

**Fig. 5 IP6 synergises with cisplatin enhancing its cytotoxic activity in BMI1$^{High}$;CHD7$^{Low}$ MB cells in vitro and improving survival in BMI1$^{High}$;CHD7$^{Low}$ MB xenografts. a** Cell viability assays of MB cells without (CTRL, shades of green, left panels) or with (BMI1$^{High}$;CHD7$^{Low}$, shades of red, right panels) the signature upon 72 h of treatment with increasing concentrations of cisplatin (CISP) or etoposide (ETOP). Histograms represent the percentages of viable cells relative to non-treated cells. $n = 6$ biologically independent experiments, one-way ANOVA. **b** Cell viability assays of control (left) or BMI1$^{High}$; CHD7$^{Low}$ (middle) MB cells upon 72 h of combination treatment with increasing concentrations of cisplatin and IP6. Histograms represent the percentages of viable cells relative to non-treated cells. $n = 6$ biologically independent experiments, two-way ANOVA. Surface plots (right panels) of synergy scores obtained after combination treatment of IP6 and cisplatin. $n = 3$ biologically independent experiments. **c** Cell viability assays of control (left) or BMI1$^{High}$; CHD7$^{Low}$ (middle) MB cells upon 72 h of combination treatment with increasing concentrations of etoposide and IP6. Histograms represent the percentages of viable cells relative to non-treated cells. $n = 6$ biologically independent experiments, two-way ANOVA. Surface plots (right panels) of synergy scores obtained after combination treatment of IP6 and etoposide. $n = 3$ biologically independent experiments. **d** Experimental design to assess the combinatorial effect of IP6 and cisplatin treatment administered in drinking water or through intraperitoneal injection (i.p.), respectively. **e, f** Kaplan–Meier survival curves of mice orthotopically xenografted with BMI1$^{High}$;CHD7$^{Low}$ (**e**) or control (**f**) MB cells treated with vehicle, IP6, cisplatin or both (IP6 + cisplatin). $n = 6$ biological independent animals per cisplatin-treated group, $n = 7$ biological independent animals per IP6 + cisplatin-treated groups, $n = 8$ biological independent animals per CTRL IP6-treated group, $n = 9$ biological independent animals per BMI1$^{High}$;CHD7$^{Low}$ IP6-treated group, $n = 7$ biological independent animals per CTRL vehicle-treated group, $n = 9$ biological independent animals per BMI1$^{High}$;CHD7$^{Low}$ vehicle-treated group, two-tailed $P$ values determined by log-rank test. **g** Quantification of the area of the tumours developed in mice treated as described in (**e**) and (**f**). $n = 8$ biological independent animals per BMI1$^{High}$;CHD7$^{Low}$ vehicle-treated group, $n = 6$ biological independent animals per BMI1$^{High}$;CHD7$^{Low}$ IP6 + Cisp-treated group, $n = 2$ biological independent animals per CTRL vehicle-treated group, $n = 4$ biological independent animals per CTRL IP6 + Cisp-treated group, one-way ANOVA. **h, i** Quantification of fractions of Ki-67 (**h**) or cleaved Caspase-3 (**i**) positive cells of the total number of tumour cells in mice treated as described in (**e**) and (**f**). $n = 5$ biological independent animals per BMI1$^{High}$;CHD7$^{Low}$ vehicle-treated group, $n = 4$ biological independent animals per BMI1$^{High}$;CHD7$^{Low}$ IP6 + Cisp-treated group, $n = 2$ biological independent animals per CTRL vehicle- and IP6 + Cisp-treated groups, one-way ANOVA. All graphs report mean ± SEM. $P$ values: *$P < 0.05$, **$P < 0.01$, ***$P < 0.001$ or ****$P < 0.0001$. Scale bars = 100 μm. Source data are provided as a Source Data file.

spectrometry. We confirmed an enrichment in MAPK/ERK signalling, as expected[10]. However, we found that proteins regulated by phosphorylation in BMI1$^{High}$;CHD7$^{Low}$ MB cells are significantly enriched for other RTK-related pathways as well, with activation of MET and FLT3 signalling being particularly highlighted. MET receptor has been described as a putative MB target[63,64], although FLT3 has never been linked to MB before. MET and FLT3 are mTOR activators[39,65,66]. The role of mTOR in SHH canonical signalling is known[67–71], hence inhibition of mTOR has been proposed as a therapeutic target for this subgroup, however, a potential role of mTOR in G4 MB is less well characterised. Transcriptome and methylome characterisation of G4γ subtype predicted activation of the PI3K/AKT/mTOR signalling pathway in this subgroup[4], and integrative proteogenomics analysis showed that RTK and downstream signalling constitute central oncogenic drivers in G4 MB[62]. Here we demonstrate that BMI1$^{High}$;CHD7$^{Low}$ specifically enhances mTOR activation in MB and not in hNSC, a finding which is in agreement with MET and FLT3 not being deregulated in these cells, which are also not impacted by inhibition of FLT3. In the neoplastic context, our data support a specific impact on mTORC1 signalling. Indeed, while Torin, a broad inhibitor of both mTORC1 and mTORC2, partially affected the viability of MB cells without the signature, treatment with the mTORC1-specific inhibitor rapamycin only impacted on BMI1$^{High}$; CHD7$^{Low}$ MB cells.

Both mTOR signalling and inositol contribute to the modulation of energy production[42–45] which is crucial for the tumour to functionally adapt to a high proliferative state. We show a neoplastic-specific energetic adaptation in a BMI1$^{High}$;CHD7$^{Low}$ context, with decreased mitochondrial function and concomitantly increased glycolytic activity, leading to increased ATP production in basal condition. This is a well-characterised phenomenon, known as the Warburg effect, which allows tumour cells to enhance aerobic glycolysis and use glycolytic intermediates to support macromolecule synthesis thereby sustaining proliferation in hypoxic condition[20,72]. Because both BMI1 and CHD7 have been previously linked to hypoxia, with BMI1 expression specifically found in the hypoxic region in glioblastoma[73] and CHD7 being downregulated in response to

low oxygen condition[74], it is conceivable that a BMI1$^{High}$; CHD7$^{Low}$ signature identifies a proportion of G4 MB with a specific advantage in hypoxic conditions due to the metabolic adaptation we have described. Notably, we observed enhanced expression of genes coding for enzymes involved in glucose metabolism and increased levels of valine and leucine by MRS in BMI1$^{High}$;CHD7$^{Low}$ G4 MB tissues confirming our conclusions from the in vitro data and raising the possibility that imaging biomarkers could be developed to identify patients with the signature. Interestingly, we show that the same signature in hNSC does not elicit the same effect, despite the increased proliferative activity acquired by the cells, raising the possibility that additional events leading to the acquisition of metabolic adaptation are essential for neoplastic transformation.

IP6 antitumour activity has been documented in different types of tumour cell lines and is mainly mediated by cell growth inhibition[27,75]; however, its role in MB had never been investigated. We show that IP6 treatment counteracted both mTOR activation and metabolic adaptation in BMI1$^{High}$;CHD7$^{Low}$ MB cells and inhibited their proliferation. Importantly, IP6 sensitivity in BMI1$^{High}$;CHD7$^{Low}$ MB cells is mediated by a novel BMI-dependent epigenetic regulation at promoters of genes involved in PLC activation, and thus lipid inositol signalling, where co-occupancy by BMI1 and H3K27me3 is demonstrated specifically in BMI1$^{High}$;CHD7$^{Low}$ MB cells. These data are in agreement with a previous observation in embryonic stem cells where genes related to inositol-phosphate metabolism were identified as PRC targets[76]. Interestingly, we also found that BMI1 co-localises at bivalent domains, marked H3K27me3 and H3K4me3, in MB cells and in particular onto the promoter of the PPIP5K2 gene, a kinase responsible for phosphorylation of IP6 to higher inositol products, which is specifically downregulated in BMI1$^{High}$; CHD7$^{Low}$ MB cells upon inositol treatment.

Our data raise the possibility that a BMI1-mediated regulation at the PPIP5K2 promoter allows BMI1$^{High}$;CHD7$^{Low}$ cells to reduce its expression upon dysregulation of inositol metabolism, therefore preventing IP6 conversion and enhancing its cytotoxic effect. In keeping with this interpretation, silencing of PPIP5K2 led to decreased cell proliferation in colon cancer cells[77]. Future studies will be required to elucidate the role of PPIP5K2 in MB

metabolism and growth and to investigate its potential role as a predictive biomarker, once its methylation status in BMI1^High; CHD7^Low G4 MB patients will be tested; something not possible in publicly available datasets as the gene is not present in the DNA methylation arrays used for these studies. Moreover, we show that MET and FLT3 promoters are differentially bound by BMI1 upon CHD7 silencing, in agreement with loss of BMI1-dependent repression of these RTKs in BMI1^High;CHD7^Low MB cells that in turn become more sensitive to their inhibition and show enhanced mTOR pathway activation.

Cancer treatment frequently benefits from a combination therapy, which relies on targeting multiple pathways to obtain a cooperative and superior effect while reducing drug resistance[78]. Adjuvant inositol treatment has been shown to attenuate the side effects of chemotherapy in breast and colon cancer[79]. We reasoned that combination therapy of cisplatin and IP6 could be effective in BMI1^High;CHD7^Low G4 MB because of the over-activation of the mTOR pathway upon cisplatin treatment[80,81], which can be counteracted by IP6 and the enhancement of cisplatin-mediated antitumoral effect by inhibition of glycolysis[82,83], which can also be induced by IP6. Our results show that IP6 treatment acts synergistically with cisplatin in BMI1^High;CHD7^Low MB cells both in vitro and in an in vivo xenograft model and suggest that IP6 administration could be implemented as adjuvant therapy in G4 MB patients that are currently treated with cisplatin[5]. Moreover, we did not observe toxicity of IP6 on hNSC or side effects upon administration to mice, in keeping with previous reports where normal bone marrow progenitor cells[84] and peripheral blood mononuclear cells and T-cell colony-forming cells are unaffected by inositol compound treatment[27].

In conclusion, deconvolution of the molecular convergence of the chromatin remodellers BMI1 and CHD7 in MB, uncovered a novel epigenetic regulation of phosphoinositol metabolism, which mediates a metabolic energetic adaptation in BMI1^High;CHD7^Low subgroup of G4 MB and predicts response to inositol pathway modulation in combination with chemotherapeutics currently used to treat patients with MB. We, therefore, identify a potentially actionable vulnerability of this particular group of patients.

## Methods

**Cell culture conditions**. ICb1299 patient-derived MB lines were obtained from Dr Xiao-Nan Li, Baylor College of Medicine, Texas Children Cancer Centre, USA[85]. ICb1299 was cultured in DMEM (high glucose, GlutaMAX™, ThermoFisher) supplemented with 10% foetal bovine serum (FBS, Gibco) and 1% penicillin–streptomycin (Gibco). CHLA-01-Med MB cells were purchased from ATCC (CRL3021) and cultured as described previously[86]. Briefly, cells were grown in DMEM-F12 (Gibco) supplemented with B27 (Gibco), 20 ng/mL human epidermal growth factor (EGF, Peprotech) and 10 ng/mL human basic fibroblast growth factor (FGF, Peprotech). MB subtype identity for the two cell lines has been previously shown[10,85].

Human foetal NSC lines were obtained from the Cancer Research UK-funded Glioma Cellular Genetics Resource (www.gcgr.org.uk) and cultured as previously described[87]. Informed consent was obtained from all donors of tissue for NSC derivation. Ethical approval was obtained from South East Scotland REC (Reference 08/S1101/1 and Amendment 08/S1101/1/AM06/1) and management approval was obtained from Lothian NHS Board. Cells were maintained at 37 °C and were sub-cultured every 3 days once they reached confluence.

**Cell viability, growth and apoptosis assays**. For growth curve and cell viability assays, MB cells or hNSCs were seeded in 24-well plates at the same density. Cells were treated for 72 h with mTOR (rapamycin, Sigma; Torin1, Calbiochem), FLT3 (TCS359, SelleckChem) or MET (PHA665752, Tocris) inhibitors or with inositol hexakisphosphate (IP6, Sigma) at the indicated concentrations. At specific time points or after appropriate treatment, cells were harvested, and the number of viable cells was counted with a haemocytometer and Trypan Blue staining or with CyQUANT Direct Red Cell Proliferation Assay Kit (ThermoFisher Scientific). Syenrgy/antagonist effect of the combined treatment was identified by Loewe Model with Combenefit software[88]. Apoptosis was assessed with Caspase-3 Colorimetric Assay kit (ab39401, Abcam) following the manufacturer's protocol. Briefly, cells treated with increasing concentrations of IP6 were lysed, and 50 μg of

protein lysate was incubated with DEVD-AFC substrate and reaction buffer for 2 h at 37 °C.

**Production of shRNA lentiviral vector and gene silencing**. GIPZ lentiviral shRNA vectors containing a hairpin sequence targeting BMI1 or CHD7 and the coding sequences for GFP and puromycin-resistance gene were purchased from Dharmacon, UK. Packaging, virus production and determination of titre were carried out as previously reported[10]. Cells were infected overnight at a multiplicity of infection (MOI) of 1. After 96 h from the infection, puromycin selection at a concentration of 2.5 μg/ml was applied to enrich the transduced population. The efficacy of gene silencing was assessed by qRT-PCR or western blot analysis.

**RNA extraction and qRT-PCR analysis**. The total RNA was isolated from cell pellets with RNeasy Micro purification kit (Qiagen) and digested with DNaseI (Applied Biosystems). The cDNA synthesis was carried out with SuperScript III Reverse Transcriptase Kit (Invitrogen) following the manufacturer's protocol. Analysis of gene expression was performed with the Applied Biosystems 7500 Real-Time PCR System using TaqMan gene expression MasterMix (Applied Biosystems) and SYBR Green PCR MasterMix (Applied Biosystems) according to standard protocols. Technical triplicates for each sample were analysed. The Ct values of all the genes analysed were normalised to the Ct of GAPDH for TaqMan probes or average Ct of ACTB and ATP5F1B for SYBR Green PCR and fold changes were calculated. Target genes and TaqMan probes used are the following: EOMES, Hs00172872_m1; MT-ND1, Hs02596873_s1; GAPDH, Hs02758991_g1. Primers used in SYBR Green qPCR are listed in Supplementary Table 1.

**Western blot analysis**. MB cells or hNSCs were lysed for 30 min on ice using RIPA lysis buffer supplemented with 2 mM PMSF, 1 mM sodium orthovanadate and protease inhibitor cocktail (PIC) (RIPA Lysis Buffer System, Santa Cruz) followed by three pulses of sonication to obtain total protein lysate. Nuclear and cytoplasmic fractions of cells were obtained with two different lysis buffers as described previously[89]. Briefly, cells were harvested in Buffer A (10 mM HEPES, pH 7.9, 10 mM KCl, 0.1 mM EDTA, 0.15% Nonidet P40 (NP40) and 0.1 mM EGTA) supplemented with 1 mM DTT and PIC. Cells were homogenised through a 26-G needle, nuclei were isolated by centrifugation and the supernatant (cyto-plasmic fraction) was collected. Nuclei were washed quickly with ice-cold PBS, suspended in Buffer B (20 mM HEPES, pH 7.9, 400 mM NaCl, 1 mM EDTA, 1 mM EGTA and 0,5% NP40) supplemented with 1 mM DTT and PIC and lysed by sonication. Protein concentration was determined using BCA Protein Assay Kit (Pierce). Equal amounts of protein were separated by SDS-PAGE and transferred onto a nitrocellulose membrane (Amersham). After transfer, the membrane was blocked for an hour at room temperature in 5% skimmed milk in TBST buffer (25 mM Tris-HCl, 137 mM NaCl and 0.1% Tween 20, pH 7.5) and probed with different antibodies. Incubation with primary antibody was performed overnight at 4 °C followed by appropriate secondary HRP-conjugated antibodies (anti-rabbit IgG or anti-mouse IgG, 1:5000, Amersham) for one hour at room temperature. Enhanced chemiluminescence (ECL Plus, Amersham) was used for the detection of the bands. The following primary antibodies were used: mouse monoclonal anti-BMI1 (1:1000, clone AF27, Active Motif), anti-GAPDH (1:1000, G8795, Sigma) and anti-Vinculin (1:5000, V4505, Sigma), rabbit monoclonal anti-phospho-RPS6 (Ser240/244, 1:1000, D68F8, Cell Signaling), anti-4EBP1 (1:1000, 53H11, Cell Signaling), anti-phospho-4EBP1 (Thr37/46, 1:1000, 236B4, Cell Signaling), rabbit polyclonal anti-CHD7 (1:500, ab117522, Abcam), anti-RPS6 (1:1000, 5G10, Cell Signaling), anti-PPIP5K2 (1:500, 16836, Novus Biological) and goat polyclonal anti-Lamin B (1:5000, sc-6216, Santa Cruz).

**Mytotracker and immunofluorescence analysis**. MB or hNSC cells, incubated for 1 h at 37 °C with 0.02 nM MitoTracker Red CMXRos (ThermoFisher Scientific), were fixed using 4% PFA, permeabilised with 0.2% Triton® X-100 and mounted for immunofluorescence analysis. Relative red fluorescent signal from mitotracker was quantified with ImageJ.

**OCR and ECAR analysis**. Metabolic extracellular flux was measured using Seahorse Extracellular Flux Analyzer XF-24 (Agilent Technology) as previously described[90]. Briefly, $1.7 \times 10^5$ MB or hNSC cells were seeded on XF-24 cell culture microplates, coated with 3.4 mg/mL BD Cell-TakTM tissue adhesive solution (BD Bioscience 354240), on the same day of the analysis. For OCR and ECAR analysis after IP6 treatment, cells were incubated with IP6 for 24 h before seeding on coated wells. Microplates containing the cell suspension were centrifuged at 180×g for 2 min and then incubated at 37 °C with unbuffered DMEM (Sigma) in a non-$CO_2$ incubator for at least 2 h. OCR was determined following sequential treatment with the ATPase inhibitor oligomycin (5 μM), the uncoupling agent FCCP (4 μM) and the electron-transport-chain inhibitors rotenone (10 μM) and antimycin A (10 μM). ECAR was measured in response to glucose (10 mM), oligomycin (5 μM) and 2DG (50 mM) treatment.

**Total ATP and ADP/ATP ratio measurements**. $1 \times 10^5$ MB or hNSC cells were cultured for 24 h and then harvested and washed with ice-cold PBS to remove

extracellular ATP. The cell pellet was resuspended in ice-cold water and sonicated for 5 min with cycles of 30 s on and 30 s off. 25 μl of cell lysate were incubated with 25 μl of CellTiter-Glo Luminescent Cell Viability Assay solution (Promega) for 10 min. ADP/ATP ratio was measured with ADP/ATP Ratio Assay Kit (Sigma Aldrich) following the manufacturer's protocol. The luminescence intensity was measured using a Synergy HT Multi-Mode Microplate Reader, normalised on the total amount of proteins and expressed as RLU (relative luminescence unit)/μg of proteins.

**Mass spectrometry sample preparation and LC–MS/MS analysis**. Protein extraction and trypsin digestion were performed as previously described[91,92]. Peptide solutions were desalted using 10 mg of OASIS-HLB cartridges (Waters, UK) and peptides were eluted with glycolic acid buffer (1 M glycolic acid, 50% ACN, 5% TFA). To enrich phosphopeptides, sample volumes were normalised using glycolic acid buffer (1 M glycolic acid, 80% ACN, 5% TFA) and TiO2 beads ((50% slurry in 1% TFA), Hichrom).

For LC–MS/MS analysis, phosphopeptides were reconstituted in 13 μL of reconstitution (97% $H_2O$, 3% ACN and 0.1% TFA, 50 fmol/μl enolase peptide digest) and sonicated for 2 min at RT. Phosphopeptides were analysed by nanoflow ultimate 3000 RSL nano instrument that was coupled online to a Q-Exactive plus mass spectrometer (ThermoFisher Scientific). Gradient elution was from 3 to 35% buffer B in 120 min at a flow rate of 250 nL/min with buffer A being used to balance the mobile phase (buffer A was 0.1% formic acid in water and B was 0.1% formic acid in ACN). The mass spectrometer was controlled by Xcalibur software (version 4.0) and operated in the positive mode. The spray voltage was 1.95 kV and the capillary temperature was set to 255 °C. The Q-Exactive plus was operated in data-dependent mode with one-survey MS scan followed by 15 MS/MS scans. The full scans were acquired in the mass analyser at 375–1500 m/z with a resolution of 70,000, and the MS/MS scans were obtained with a resolution of 17,500.

**Peptide identification and quantification**. For peptide identification, MS raw files were converted into Mascot Generic Format using Mascot Distiller (version 2.7.1) and searched against the Swiss-Prot database (release September 2019) (https://www.uniprot.org/statistics/Swiss-Prot) restricted to human entries using the Mascot search daemon (Ref 2, version 2.6.0) with a FDR of ~1% and restricted to the human entries. Allowed mass windows were 10 ppm and 25 mmu for parent and fragment mass to charge values, respectively. Variable modifications included in searches were oxidation of methionine, pyro-glu (N-term) and phosphorylation of serine, threonine and tyrosine. Peptides with an expectation value <0.05 were considered for further analysis.

The mascot result (DAT) files were extracted into excel files. For peptide quantification, Pescal Software [3] was used to construct extracted ion chromatograms (XIC) for all the identified peptides across all conditions and calculating the peak heights. These peptide peak heights were then normalised to the sum of the intensities for each individual sample and the average fold change between conditions could be determined. Statistical significance between conditions was considered significant when the Student's $T$ tests produced $P < 0.05$ following Benjamini–Hochberg (BH) multiple testing correction. Kinase substrate enrichment analysis (KSEA) was performed as described before[93]. Briefly, peptides differentially phosphorylated between a set of samples (at non-adjusted $P < 0.05$) were grouped into substrate sets known to be phosphorylated by a specific kinase as annotated in the PhosphoSite (https://www.phosphosite.org/), Phospho.ELM (http://phospho.elm.eu.org/) and PhosphoPOINT (http://kinase.bioinformatics.tw/) databases[94–97]. To infer the enrichment of substrate groups across sets of samples, the hypergeometric test was used, followed by BH multiple testing correction.

**DNA methylation analysis**. Total genomic DNA was extracted with RNA/DNA/Protein Purification Plus kit (Norgen), and DNA methylation data were assayed on the Illumina Human Methylation EPIC microarray after bisulfite conversion. Raw data were preprocessed using the Bioconductor ChAMP package in R to remove failed detections and probes with design flaws[98]. Normalisation was performed using the SWAN algorithm[99]. Probes were mapped to genes using the methylation array annotation. One probe may map to multiple genes. These operations were carried out on both cell lines simultaneously, after which they were analysed separately. Two preprocessing steps were then applied to increase our ability to detect differential methylation. First, any probes whose $M$ values had a range less than 1 M value unit across all experimental groups were removed, then any probes whose $M$ values varied by greater than 5 M value units within a single experimental group were removed. Differentially methylated probes were identified using the limma package in R[100]. Probes with a false discovery rate <0.05 were declared differentially methylated.

**RNA sequencing (RNA-Seq) analysis**. The total RNA was extracted with RNA/DNA/Protein Purification Plus kit (Norgen), polyA mRNA was selected and library-prepared using the NEBNext Ultra II Directional RNA Library Prep kit. Libraries were size-selected, multiplexed and 75-bp paired-end sequences were obtained with HiSeq4000 (Illumina). Quality control was performed with FastQC and adapter sequences were removed using Trimgalore v0.6.5 (www.bioinformatics.babraham.ac.uk/projects/). Reads were then aligned to the Ensembl

GRCh38 human reference genome using STAR v2.6.1[101], with gene count quantification mode, retaining only uniquely mapped reads. Lowly expressed genes (CPM < 1) were filtered out via a custom-made script in R, and further biotype filtering was performed in R with the Bioconductor packages NOISeq[102] and biomaRt[103], removing residual highly expressed mitochondrial and ribosomal RNA. Trimmed mean of M-values (TMM) normalisation was applied to the dataset prior to differential expression (DE) analysis, performed using the Bioconductor package edgeR[104]. Specifically, a negative binomial generalised log-linear model (glmfit) was fitted to the read counts and the likelihood ratio test (glmLRT) was conducted for each comparison of interest. The Benjamini–Hochberg FDR cut-off was set at 0.05. The versions of all relevant Bioconductor packages were compatible with R v3.5.3.

**Chromatin immunoprecipitation (ChIP) assays**. BMI1, H3K4me3 and H3K27me3 ChIPs were performed using ChIP-IT High Sensitivity kit (Active Motif) following the manufacturer's instructions. Briefly, $1 \times 10^6$ cells were fixed with the formaldehyde-based fixing solution for 15 min at room temperature and lysed with provided lysis solution supplemented with protease inhibitors. Next, nuclei pellets were lysed and chromatin sonicated with Bioruptor Plus sonication device (Diagenode) to obtain DNA fragments within the recommended 200–1200-bp range. In total, 25 μg or 10 μg of sheared chromatin was then incubated with 4 μg of antibody against BMI1 (BMI1, clone AF27, Active Motif), H3K27me3 (Diagenode) or H3K4me3 (Diagenode) overnight at 4 °C with rotation. Following incubation with Protein G agarose beads, bound chromatin was washed, eluted and purified following the manufacturer's protocols. Validation by qPCR-ChIP on target genes was done before proceeding to sequencing. ChIPed DNA was end-repaired, A-tailed and adapter-ligated before size selection and amplification. The obtained libraries were QC'ed and multiplexed before 75-bp paired-end sequencing on HiSeq4000 (Illumina).

**ChIP-Seq analysis**. The quality of ChIP-Seq samples was first assessed via FastQC and TrimGalore, removing low-quality and adapter sequences. The average Phred score of the surviving reads across all samples was 30 and the average sequencing depth was 36.3 M (min = 22.1 M, max = 56.7 M). The alignment to the Ensembl GRCh38 human reference genome was performed via Bowtie v2.3.4[105] with default parameters, in concomitance with the usage of samtools[106] for the post processing and sorting of the Binary Alignment Map (bam) files. Exploratory tools such as deeptools[107], plotCorrelation, plotPCA and plotFingerprint on Python v2.7.15 were used to further assess sample characteristics and to address the potential presence of outliers. After performing post-alignment quality checks, peaks were called via the MACS2 algorithm (subroutine *callpeak*)[108] against the corresponding input background. The shifting model was disabled to make different datasets comparable and the "--broad" option was enabled for the analysis of the histone mark H3K27me3. A minimum fold enrichment of 2 was selected with an FDR of 0.05, in both narrow peak and broad peak (--broad-cut-off) statistical analyses. The Bioconductor packages in R GenomicRanges[109] and ChIPSeeker[110] were used to find regions of consensus peaks between the two cell lines, for each antibody/condition, and to annotate them based on the location with respect to the nearest transcription start site (TSS): promoters (within 3 Kb from the TSS), exons, introns, 5′ UTR, 3′ UTR and distal intergenic. The latter was merged with so-called downstream regions. The versions of all relevant Bioconductor packages were compatible with R v3.5.3.

**ChIP-Seq heatmap generation**. The deeptools algorithms bamCompare, bigWigCompare and plotHeatmap were used to produce relevant bigwig files and heatmaps, in order to assess region-wide and genome-wide coverage. Specifically, coverage tracks were first obtained by normalising each sample alignment file against the corresponding input (--operation log2ratio) and then pooled at both replicate and cell line levels (--operation mean), to obtain a representative sample for each antibody/condition. The regions chosen to be visualised in the heatmaps were those of consensus peaks (details in the figure legends), centred on their nearest TSS.

**Pathway and gene set analysis**. Ingenuity pathway analysis (IPA, Qiagen), Reactome (https://reactome.org/)[111] and Panther Gene Ontology tool (http://geneontology.org/) were used to assess biological pathways and gene ontology terms that showed significant enrichment in the various gene sets. Functional analysis of significant DE and DMP gene lists obtained for MB cells or tumour samples was carried out with IPA software with canonical pathways tool following the manufacturer's standard protocol. Pathways that were significant in both RNA-Seq and DNA methylation analysis and conserved between cells and tumour samples were visualised by Venn diagrams. Reactome Pathway Database was used to interrogate datasets of differentially phosphorylated proteins, while Panther Gene Ontology was used to analyse GO terms significantly enriched in ChIP-Seq analysis. The enrichment for each term was deemed statistically significant if the adjusted $P$ value (FDR) was lower than 0.05.

Cytoscape v. 3.7.2[112] was used to visualise relevant biological networks of enriched pathways, together with EnrichmentMap and AutoAnnotate applications.

Several layout parameters were tuned to achieve the current Cytoscape visualisation.

**Genetically engineered mouse model**. *Math1Cre;STOPFloxBmi1;Chd7f/+*, *Math1Cre;STOPFloxBmi1 and STOPFloxBmi and Chd7f/+* control mice were generated by crossing *Math1Cre* mice or *Math1Cre;Chd7f/f* mice with *STOP-FloxBmi1* mice. Mouse lines were as described previously[113–115]. In vivo BrdU labelling and analyses were performed as described previously[114]. For immuno-stainings, brains were dissected, fixed in 4% PFA overnight and processed for immunostaining as previously described[10], using rabbit anti-EOMES (Sigma Aldrich).

**Animal experiments**. All procedures were performed in accordance with licences held under the UK Animals (Scientific Procedures) Act 1986 and later modifica-tions and conforming to all relevant guidelines and regulations. Mice were kept at ~18–23 °C with 40–60% humidity in a 12-h light/12-h dark cycle. Required sample sizes were calculated by an a priori power analysis and mice were assigned ran-domly to the different groups of treatment. $1 \times 10^5$ MB cells were injected into the right cerebellar hemisphere of P5 mice. After 2 weeks from the injection, mice were treated with cisplatin and/or IP6. Cisplatin (Sigma) was administered by intra-peritoneal (i.p.) injection at 3.5 mg/kg in saline solution. 2% IP6 (Sigma) was administered in sterile drinking water, pH-adjusted to 7.4 for oral administration and replaced every 48 h. Control mice were i.p.-injected with saline solution, as the vehicle.

**Data analysis and statistical methods**. The number of experiments (*n*) used for all the statistical tests is specified in each figure legend. All quantitative data are presented as mean ± standard error of the mean (SEM) of at least three experi-ments, unless otherwise specified. The statistical significance was determined by two-tailed unpaired Student's *t* test, one-way ANOVA followed by Tukey's multiple-comparison test or two-way ANOVA followed by Dunnet's multiple-comparison test. Survival of xenografted mice was estimated with Kaplan–Meier survival analysis, and significance was determined with Log-rank (Mantel–Cox) test. Statistical analysis was performed using GraphPad software and statistical significance is represented as $*P < 0.05$, $**P < 0.01$, $***P < 0.001$ or $****P < 0.0001$.

**Reporting summary**. Further information on research design is available in the Nature Research Reporting Summary linked to this article.

## Data availability
The authors declare that all the data supporting the findings of this study are available within the article and its Supplementary Information files. The datasets generated in this study and processed data are available in the NCBI Gene Expression Omnibus database (GSE156077) or are available from the corresponding author upon reasonable request. Publicly available datasets used in the study: GSE85217 (Expression data from primary medulloblastoma samples), Reactome (https://reactome.org/), Panther Gene Ontology tool (http://geneontology.org/), Swiss-Prot (https://www.uniprot.org/statistics/Swiss-Prot), PhosphoSite (https://www.phosphosite.org/), Phospho.ELM (http://phospho.elm.eu.org/) and PhosphoPOINT (http://kinase.bioinformatics.tw/). Source data are provided with this paper.

## Code availability
The custom-made code used in this study is available in the GitHub repository (https://github.com/nickpom88/rnaseq/blob/main/filter_genes_np2.R).

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

## Acknowledgements

We are grateful to the BSU staff for help in the daily care of our mouse colony and to the Blizard Core Facilities (Imaging and FACS) for technical advice. Access to medullo-blastoma tissue was obtained through the CCLG tissue bank (project number 2015 BS 05). This work is supported by grants of the Medical Research Council UK (MR/N000528/1) to S.M. and M.A.B. and a Brain Tumour Research Centre of Excellence award to S.M. Y.M.L. is funded by a Cancer Research UK Accelerator grant Cl 15121A 20256. Funding from Children with Cancer UK (15/188), Action Medical Research and The Brain Tumour Charity (GN2181) to C.D.B., S.C.C. and A.P. are also acknowledged. A.P. was funded by an NIHR Research Professorship (NIHR-RP-R2-12-019). S.Br. is partly supported by the Department of Health's NIHR Biomedical Research Centre's funding scheme to University College London Hospitals.

## Author contributions

S.Ba. and S.M. conceived the project, S.Ba., M.A.B. and S.M. designed the experiments, S.Ba. performed and analysed all the experiments, X.Z. and J.W. contributed to the animal experiments, M.V.N.C. co-designed the metabolic assays, N.P. analysed ChIP-Seq data, G.R. analysed DNA methylation and RNA-Seq data, Y.M.L. and S.Br. performed image analysis to quantify IHC results, G.M. and S.M.P. provided human foetal neural stem cells, C.D.B., S.C.C. and A.P. conducted ex vivo MRS experiments and S.Ba. and S.M. wrote the paper with contributions from all authors.

## Competing interests

The authors declare no competing interests.
