## [Peer Review File · Nature Communications]

Reviewers' Comments:

Reviewer #1:

Remarks to the Author:

The manuscript presents data on inositol playing a specific role in G4 medulloblastoma. The most impactful finding is that IP6 produces a significant anti-tumor effect in a xenograft model of the subset of G4 medulloblastomas with BMI1High;CHD7Low signature. This effect seems to be specific to this subgroup, as a control G4 xenograft does not have the same sensitivity to IP6. In theory, this finding could lead to new ways of targeting a subset of G4 medulloblastoma patients if the correct subset could be identified. There are weaknesses in the work that detract from the potential impact and from some of the conclusions drawn, and these weaknesses would need to be addressed before the paper is ready for publication.

The data in Fig 1 is not well integrated into the rest of the work. In this figure, the authors test the effect for an oncogenic effect of genetically engineering conditional overexpression of BMI1 and heterozygous deletion of CHD7 in Math1-expressing cerebellar progenitors. They do not find an oncogenic effect, but seek to conclude that this genetic manipulation alters the fate of Math1+ progenitors toward a UBC phenotype. While this finding is of interest, it is not essential to the rest of the manuscript, and the conclusions of the manuscript would not be altered without this figure. As a result, it seems misplaced in this manuscript. If the authors had performed experiments on the Math1-lineage cells with conditional overexpression of BMI1 and heterozygous deletion of CHD7 in the rest of the paper, Figure 1 would be relevant, but the rest of the paper does not concern these cells or use these mice.

The rest of the paper addresses observations of G4 cell lines and G4 patient samples. More detail is needed describing the cell lines. The authors state that they are using "G4 MB cell lines upon modelling of the signature as previously described", and reference a prior publication. This reference is simply not enough information for the reader. When the authors compare "samples with and without the signature" the reader needs to know what cell lines were compared and how these cell lines were selected and maintained. Reading the prior publication does not provide this information. The discussion text should be self-contained enough to provide essential information required to understand the study.

The failure to specify exactly what cell lines are used detracts from the reader's ability to interpret data throughout the text. When cell lines with high BMI1/low CHD7 cells are compared with control G4 cell lines, understanding the comparison requires information on selection and possible genetic manipulation. This information may be relevant to the observed decreased mitochondrial function in the BMI1/low CHD7 cell line. Another example is the xenograft experiment, in which "BMI1High;CHD7Low and control G4 MB cells" are compared. Which of these two groups truly models G4 medulloblastoma- do G4 cell lines that do not show the BMI1High;CHD7Low signature have other alterations that allow their growth, and do these alterations detract from their relevance? The reader cannot address this question or understand the implications of the data, without more information on the model.

In addition to these structural issues, there are several specific issues that need to be addressed:

The statement "Aerobic glycolysis provides up to 60% of ATP 80 production in MB" is not supported by reference 24, which does not refer specifically to medulloblastoma.

Figure 1C does not add support to the idea that CGNs are reduced in the Math1Cre;STOPFloxBmi1;Chd7f/+ mice. There is no reason to expect that UBCs do not express NeuN. If the team found that Eomes+ neurons are NeuN-, they should show this finding as a figure. Without positive information proving that UBCs do not express NeuN, it is reasonable to

assume that UBCs, as neurons do express NeuN. As a result, the cell counts in Fig 1C do not show that CGNs are decreased, but rather that the overall neuronal population is decreased.

Figure 1D- the authors need to show more documentation to use PAX3 as a UBC marker. Published scRNA-seq data show PAX3 as a marker for interneurons outside the Math1 lineage. The use of PAX3 as a UBC marker should be better substantiated or dropped.

The results depicted in Figure 2 make the case that an unsupervised analysis of gene expression and DNA methylation identified inositol metabolism as the only "common and conserved pathway". It is not clear whether the analysis required tuning to produce this singular result, or whether another observer, starting from the same data would invariably reach the same conclusion. To maintain rigor and reproducibility, and to allow the reader to understand what specific analytic steps were performed, more details of the pathway analysis methods are needed. More detail needs to be added to the statement in the method "Several layout parameters were tuned to achieve the current visualization."

The studies of 2A-D need to make clear that what is compared is cell lines growing in culture vs tumor samples freshly resected from the in vivo environment. Metabolic conditions in culture and in vivo may be very different. By integrating samples from in vitro cell lines and in vivo tumors, the authors are seeking metabolic pathways that are consistently activated across different environmental conditions. The fact that they found conserved pathways is a strength of the work, and it should be emphasized.

2J, rapamycin concentrations are too high to be considered specific or informative. This particular panel dose not make a strong case for mTOR activation being important. A more effective mTOR inhibitor may produce a more convincing result.

2K IP6 concentrations required to elicit an effect are very high, raising the question of whether this effect is physiologically relevant. At the 1 mM concentration, the control and experimental groups do not show a statistically significant difference. Also in 2K, control and experimental groups should not be normalized to different means, but rather to a single mean, to allow for comparisons between groups at each dose level. As currently presented, the mean for untreated control and experimental is set to 1, and as a result, the untreated control and experimental cannot be compared.

Fig3 should include the full OCR tracing for BMI1High;CHD7Low G4 MB cells to show in detail the effects of IP6 treatment, as the statement that OCR is not affected is not sufficiently supported by the limited data presented in Fig S3E.

ATP studies of Fig 3C and 3G. 3C does not support the interpretation that "BMI1High;CHD7Low G4 MB cells produce significantly more ATP than controls" because ATP content is determined by both ATP production and ATP utilization. What 3C demonstrates is that steady state ATP levels are different in the two groups compared. This same issue of interpretation applies to 3G, where there is also an inconsistency in the conclusion drawn- where in 3C decreased mitochondrial function and increased glycolysis are said to cause increased ATP production, while in 3G increased mitochondrial function and increased glycolysis are said to cause reduced ATP production. The logic that the authors use needs to be more fully explained, or the interpretation needs to be revised. Since differences in ATP utilization have not been considered, and such differences could affect steady state ATP levels, and the interpretation of ATP levels seems problematic, differences in ATP utilization should be incorporated into the interpretation.

Results presented in lines 225-266 should be broken into at least two paragraphs to enhance readability.

Reviewer #3:

Remarks to the Author:

Beautiful paper that correlates BMI expression and CHD7 suppression in a subset of G4 medulloblastoma, and demonstrates resulting mTOR dependent deregulation of inositol metabolism and therapeutic inroads that result. The ability to integrate large datasets and complex data into easily understood figures was impressive. Effects sizes were robust. I had only minor suggestions.

1. For experiments in Fig 2 and going forward, the authors compare G4 lines with their signature to either G4 lines without the signature or with hNSCs. In many cases, it was unclear to me whether the green bars without signature were always hNSCs or may represent signature negative G4 lines. It was also unclear to me which experiments were done in human and which in mouse lines. Can the authors clarify these issues, and can key experiments be repeated in additional signature pos and negative human lines, to assure the generalizability of conclusions?

2. Also for experiments in Fig 2, the authors conclude that, "These results validate the in silico prediction of an enhanced signalling through the tyrosine-kinase receptors FLT3 and MET in BMI1High;CHD7Low G4 MB but not in hNSC with same molecular signature."

This isn't strictly true, they have shown increased dependence, but can they also show by westerns, that these cells show increased signaling, (and that inhibitors actually block their intended targets) as claimed?

3. Can human PDX models tested in in vivo therapy studies in fig 5?

Reviewer #4:

Remarks to the Author:

Badodi and colleagues implicate epigenetic regulation of inositol metabolism as a novel mechanism underlying Group 4 MB associated with an expression signature defined by BMI1-high;CHD7-low. A fundamental concern of the study and the majority of the experimental results is the use of in vitro cultured Icb-1299 and CHLA-01-Med cell lines as Group 4 MB cell line models, as these lines appear to be more closely related to Group 3 MB than any other subgroup. Although the authors have previously interpreted these lines as being representative of Group 4 MB (ref. 10), this interpretation is hampered by close inspection of the datasets presented in their prior report (ref. 10) and by more recent studies that have defined these models as Group 3 MB lines. Given this major concern of subgroup identity, it is challenging to appreciate the relevance of the findings presented in the current manuscript, given the uncertainties and doubt concerning biological context. If the author's intent is to implicate a novel mechanism pertinent to Group 4 MB biology, and to nominate potential treatment options for this important subgroup of patients, they are urged to revisit their analyses and validate the primary study conclusions using authentic Group 4 models (i.e. PDX) that have been substantiated as molecularly accurate and representative models of this subgroup.

1. By comparing two cultured MB cell lines (Icb1299 and CHLA-01-Med) with or without BMI1-high;CHD7-low signature, the authors examined differentially expressed genes and dysregulated pathways, and identified metabolic adaptation in BMI1-high;CHD7-low MB cell lines which leads to an impaired mitochondrial respiration and enhanced aerobic glycolysis. The authors also showed that IP6 treatment alone or in combination with cisplatin treatment can more specifically inhibit the growth of BMI1-high;CHD-low MB cell lines compared to control cells. However, there are some fundamental concerns about the use of these cell lines and their inferred identity. First, the subgroup identity of these cell lines is controversial. In one of the previous studies cited here (ref. 10), both Icb-1299 and CHLA-01-Med appear to be more similar to Group 3 MB based on gene

expression analysis and were indistinguishable from other Group 3 MB lines. Likewise, in multiple recent studies published by prominent labs (PMID: 33046443; PMID: 31341285), Icb-1299 is annotated as a Group 3 MB. Second, in vitro culture of MB cells has been shown to extensively change their gene expression and tumor biology compared with the original tumors or PDXs. Validation of the major conclusions presented in the study using more physiologically relevant models (i.e. bona fide Group 4 MB PDX) is important and required to effectively substantiate the findings reported here.

2. In Figure 1A-C, the authors implicate an increased proportion of Eomes-positive cells concomitant with a reduction in NeuN-positive cells in the developing cerebella of Math1Cre;STOPFloxBmi1;Chd7f/+ transgenic mice. Although the authors quantify these differences compared to relevant controls, the proportional differences in these populations appear to be marginal at best based on representative IHC. The authors interpret these differences as a potential lineage shift towards an expansion of the UBC lineage and reduction in the granule neuron lineage. However, these inferences are based on the IHC-derived expression patterns of two lineage markers (Eomes, NeuN). More comprehensive analyses are required to substantiate these claims. The authors should perform single-cell RNA sequencing analysis of the developing cerebella in both the control and Math1Cre;STOPFloxBmi1;Chd7f/+ transgenics to support their current findings.

3. In Figure 1G and 4A, the authors indicate that CHD7 knockdown or IP6 treatment reduced cell proliferation by examining the number of viable cells during culture. However, this method cannot distinguish proliferation vs apoptosis. The authors should perform BrdU incorporation or apoptosis assays (i.e. annexin V staining) to determine whether cell proliferation or apoptosis was affected under these experimental conditions.

4. Aerobic glycolysis has been shown to be a therapeutic target in Group 3 MB. Genes involved in aerobic glycolysis are also highly expressed in Group 3 MB compared with other subgroups (PMID: 30862721). In the current study, the authors showed enhanced aerobic glycolysis in BMI1-high;CHD7-low MB cells compared with control cells. To further confirm these findings in human patient samples, the authors should examine the expression of glycolytic genes in Group 4 MB with or without the BMI1-high;CHD7-low signature.

Response to reviewers' comments

Reviewer #1 comments (expertise: Medulloblastoma/metabolism and therapy)

The manuscript presents data on inositol playing a specific role in G4 medulloblastoma. The most impactful finding is that IP6 produces a significant anti-tumor effect in a xenograft model of the subset of G4 medulloblastomas with BMI1^{High};CHD7^{Low} signature. This effect seems to be specific to this subgroup, as a control G4 xenograft does not have the same sensitivity to IP6. In theory, this finding could lead to new ways of targeting a subset of G4 medulloblastoma patients if the correct subset could be identified. There are weaknesses in the work that detract from the potential impact and from some of the conclusions drawn, and these weaknesses would need to be addressed before the paper is ready for publication.

The data in Fig 1 is not well integrated into the rest of the work. In this figure, the authors test the effect for an oncogenic effect of genetically engineering conditional overexpression of BMI1 and heterozygous deletion of CHD7 in Math1-expressing cerebellar progenitors. They do not find an oncogenic effect, but seek to conclude that this genetic manipulation alters the fate of Math1+ progenitors toward a UBC phenotype. While this finding is of interest, it is not essential to the rest of the manuscript, and the conclusions of the manuscript would not be altered without this figure. As a result, it seems misplaced in this manuscript. If the authors had performed experiments on the Math1-lineage cells with conditional overexpression of BMI1 and heterozygous deletion of CHD7 in the rest of the paper, Figure 1 would be relevant, but the rest of the paper does not concern these cells or use these mice.

The reviewer acknowledges that modelling the BMI1^{High};CHD7^{Low} signature during cerebellar development is of interest; however they feel these experiments are not strictly necessary taking into account the main point we want to make in our manuscript. We have decided to include these data in the paper because hyperplasia of UBC progenitors, the most promising putative cell of origin of MB G4, is a strong indicator of a role for these two oncogenic events in the pathogenesis of this MB subgroup. Moreover, the lack of full blown neoplastic transformation despite ERK signalling overactivation in a non-neoplastic context¹ is what prompted us to explore other BMI1^{High};CHD7^{Low}-mediated MB-specific molecular deregulations which was essential to discover the role of IP6.

For these reasons we would be keen to keep Figure 1 to set the scene for our genome wide analysis comparing BMI1^{High};CHD7^{Low} MB cells and G4 tumour samples. Should the reviewer however, not be convinced by this explanation, we could remove the figure.

The rest of the paper addresses observations of G4 cell lines and G4 patient samples. More detail is needed describing the cell lines. The authors state that they are using "G4 MB cell lines upon modelling of the signature as previously described", and reference a prior publication. This reference is simply not enough information for the reader. When the authors compare "samples with and without the signature" the reader needs to know what cell lines were compared and how these cell lines selected and maintained. Reading the prior publication does not provide this information. The discussion text should be self-contained enough to provide essential information required to understand the study.

The failure to specify exactly what cell lines are used detracts from the readers ability to interpret data throughout the text. When cell lines with high BMI1/low CHD7 cells are compared with control G4 cell lines, understanding the comparison requires information on selection and possible genetic manipulation. This information may be relevant to the observed decreased mitochondrial function in the BMI1/low CHD7 cell line. Another example is the xenograft experiment, in which "BMI1^{High};CHD7^{Low} and control G4 MB cells" are compared. Which of these two groups truly models G4 medulloblastoma- do G4 cell lines that do not show the BMI1^{High};CHD7^{Low} signature have other alterations that allow their growth, and do these alterations detract

Professor Silvia Marino

from their relevance? The reader cannot address this question or understand the implications of the data, without more information on the model.

Further details about the MB cell lines used have now been added in the relevant section of the Results (lines 138-141).

ICb1299 and CHLA-01-Med MB cell lines show a higher expression of BMI1 compared to normal cerebellum, so they represent a suitable BMI1 overexpressing MB in which a BMI1^{High};CHD7^{Low} signature can be modelled by CHD7 silencing. They retain the G4 subgroup affiliation of the parental tumour, hence they are a suitable model for this subgroup. The advantage of this setting is that there is no confounding factor of a different genetic background or alterations, which as pointed out by the reviewer, could be difficult to disentangle from the specific role of BMI1-CHD7 in MB pathology. We are, so to say, using an isogenic control. Ideally though, we would have wanted to also use MB cell lines established from patients with the signature. However, these cells lines do not exist to our knowledge. Therefore, we have rigorously validated any finding obtained in the MB cell lines engineered to model the signature on G4 MB datasets with a BMI1^{High};CHD7^{Low} signature to ensure the findings were relevant for this subgroup of G4 MB.

In addition to these structural issues, there are several specific issues that need to be addressed:

The statement "Aerobic glycolysis provides up to 60% of ATP 80 production in MB" is not supported by reference 24, which does not refer specifically to medulloblastoma.

We believe that the findings of Moreno-Sanchez et al.² do support our statement. Specifically we refer to Table 3, reported below for convenience, in which the authors describe the contribution of glycolysis or oxidative phosphorylation to the cellular ATP supply in different types of tumours. Medulloblastoma is listed together with oligodendroglioma and meningioma as a tumour in which glycolysis has been estimated to account for 60% of ATP production.

Table 3
Energy metabolism in malignant tumor cells

Tumor	Organism	Prevailing energy pathway	% ATP contribution
Glioma C6	Rat	OxPhos	80
Oligodendroglioma, meningioma, medulloblastoma	Human	Glycolysis	60
Glioblastoma multiforme, Astrocytoma C6	Human, Rat	Glycolysis and OxPhos	50
Transformed brain	Hamster	OxPhos	71
Colon sarcoma	Human	OxPhos	70
Novikoff hepatoma	Rat	Glycolysis	75
Ehrlich Lettré, Ehrlich, Walker-256, Morris 3683 and Dunings LC18 hepatomas; ascites mouse cancer; sarcoma 27	Rat, mouse	Glycolysis and OxPhos	50
Reuber H-35, Morris (7793,7795, 7800, 5123), and AS-30D hepatomas	Rat	OxPhos	97
Lung carcinoma	Human	OxPhos	95
Breast cancer	Human	OxPhos	95
MCF7 breast carcinoma	Human	OxPhos	80
Melanoma	Human	OxPhos	97
HeLa cervix, ovarian and uterus carcinomas	Human	OxPhos	90

From Moreno-Sanchez et al, *The bioenergetic of cancer: is glycolysis the main ATP supplier in all tumour cells?*, 2009, Biofactors.

Figure 1C does not add support to the idea that CGNs are reduced in the *Math1Cre;STOPFloxBmi1;Chd7f/+* mice. There is no reason to expect that UBCs do not express NeuN. If the team found that Eomes+ neurons are NeuN-, they should show this finding as a figure. Without positive information proving that UBCs do not express NeuN, it

is reasonable to assume that UBCs, as neurons do express NeuN. As a result, the cell counts in Fig 1C do not show that CGNs are decreased, but rather that the overall neuronal population is decreased.

The point the reviewer is making here is entirely justified as we did not explicitly mention that in the adult and developing cerebellum, NeuN has been shown to be expressed in postmitotic granule neurons but not in any other cerebellar interneurons, including UBC³. This was the reason we had chosen NeuN to identify granule neurons. We are now clarifying this in the Results and citing the most relevant paper (line 112).

Figure 1D- the authors need to show more documentation to use PAX3 as a UBC marker. Published scRNA-seq data show PAX3 as a marker for interneurons outside the Math1 lineage. The use of PAX3 as a UBC marker should be better substantiated or dropped.

We have included in our analysis all 6 genes recently described as expressed in UBC⁴; however not all these genes are expressed in UBC exclusively. Some of them, such as PAX3, are expressed in both UBC and GCP progenitors, as the reviewer correctly pointed out and as described previously⁴. We have now clarified this in the relevant paragraph in the Results (lines 117-118).

The results depicted in Figure 2 make the case that an unsupervised analysis of gene expression and DNA methylation identified inositol metabolism as the only “common and conserved pathway”. It is not clear whether the analysis required tuning to produce this singular result, or whether another observer, starting from the same data would invariably reach the same conclusion. To maintain rigor and reproducibility, and to allow the reader to understand what specific analytic steps were performed, more details of the pathway analysis methods are needed. More detail needs to be added to the statement in the method “Several layout parameters were tuned to achieve the current visualization.”

We agree this is an important point to clarify and we have now expanded the description of the pathway analysis performed and added details on tools/software in the Methods section (lines 700-706).

In brief, Pathway analysis for RNAseq and DNA methylation was carried out with the Ingenuity Software Analysis (IPA, Qiagen) using the Canonical Pathways tool, inputting the four lists of significant genes (two for RNAseq and two for DNA methylation) obtained for cells and G4 MB samples with and without the BMI1^{High};CHD7^{Low} signature. Subsequently, the four lists of significant pathways were analysed and visualized with Venn diagrams to find (i) terms in common between RNAseq and DNA methylation and (ii) those conserved between cell model and tumour samples. We did not change any parameter in the IPA Canonical Pathway analysis tool which exploits the comprehensive, manually curated content of the QIAGEN Knowledge Base and any other observer could reproduce this finding.

The statement “Several layout parameters were tuned to achieve the current visualization” refers to the Cytoscape tool, where visualization parameters were adjusted to represent the bubbles in the plots with sizes proportional to number of genes of the specific term and with shades of colour based on FDR values, as specified in the figure legends.

The studies of 2A-D need to make clear that what is compared is cell lines growing in culture vs tumor samples freshly resected from the in vivo environment. Metabolic conditions in culture and in vivo may be very different. By integrating samples from in vitro cell lines and in vivo tumors, the authors are seeking metabolic pathways that are consistently activated across different environmental conditions. The fact that they found conserved pathways is a strength of the work, and it should be emphasized.

We are grateful to the reviewer for noticing this fundamental strength of our analysis and for appreciating our efforts in integrating the *in vitro* MB model and the G4 tumour samples to ensure we were focussing on clinically relevant findings. As requested, we are now emphasising this more in the Discussion (lines 377-379[SB1]).

Professor Silvia Marino

2J, rapamycin concentrations are too high to be considered specific or informative. This particular panel dose not make a strong case for mTOR activation being important. A more effective mTOR inhibitor may produce a more convincing result.

This is a mistake on our side, which has now been corrected. The molarity of the Rapamycin concentration should have been in nM. We apologise for the confusion.

Moreover to strengthen our findings we used Torin, another mTOR inhibitor, which is able to block both mTORC1 and mTORC2 (see Figure S2K); here we observe specific inhibition of viability in BMI1^{High};CHD7^{Low} MB cells (see Figure 2J left top and bottom panels).

2K IP6 concentrations required to elicit an effect are very high, raising the question of whether this effect is physiologically relevant. At the 1 mM concentration, the control and experimental groups do not show a statistically significant difference.

IP6 is a natural poly-phosphorylated carbohydrate present in cereals and legumes which can be administered at high doses without encountering toxicity. We have chosen the concentrations of IP6 to be used for *in vitro* and *in vivo* assays on the basis of several published papers (e.g. colon cancer^{5,6}, breast cancer⁷, leukaemia⁸, prostate cancer⁹, osteosarcoma¹⁰). Our study and others^{8,11} have confirmed that IP6 administration has negligible side effects on normal cells both in terms of viability (previous and current version of the manuscript in Figure 4C) and metabolism (newly added Figure S3I). Neonates treated with 2% IP6 in the drinking water (equivalent to 30mM) do not show any side effect as evaluated by body weight mass increase throughout the treatment (Figure 1, this rebuttal).

Furthermore, we estimate that 1mM IP6 is a concentration that can indeed be achieved physiologically in the brain because 1) IP6 is absorbed by rodents when dietary administered and distributed to several organs, including the brain¹²; 2) 10-fold higher concentration of IP6 is achieved in the brain compared to other tissues after dietary administration¹³ and 3) a study in human demonstrates 28% absorption of IP6 in ileostomy patients¹⁴. Hence, if the 28% of the 30mM IP6 *in vivo* administered is absorbed then it can be estimated that a physiological concentration of 8.4mM can be achieved in the body, without taking into account that IP6 concentration can be 10-fold higher in the brain than other organs.

Also in 2K, control and experimental groups should not be normalized to different means, but rather to a single mean, to allow for comparisons between groups at each dose level. As currently presented, the mean for untreated control and experimental is set to 1, and as a result, the untreated control and experimental cannot be compared. We agree with the reviewer on this point. In the previous version of the figure we arbitrary set both untreated samples as equal to 1 to control for the increased phosphorylation of RPS6 in BMI1^{High};CHD7^{Low} MB cells (Figure 2I). We were hoping to equalize this baseline difference, to then compare the effect of IP6 treatment. However, we acknowledge that to avoid misinterpretation we should have separated control and signature results in two different graphs. We now show levels of RPS6 phosphorylation for all the samples as relative to the mean of vehicle-treated BMI1^{High};CHD7^{Low} MB cells samples, as per reviewer suggestion (Figure 2K). The plot shows that IP6 treatment significantly reduces mTOR pathway activation only in BMI1^{High};CHD7^{Low} MB cells. Moreover, as the reviewer highlighted, we show that the increased activation of mTOR in BMI1^{High};CHD7^{Low} cells is reverted to the level of control cells and there is no significant difference between control cells (IP6 or vehicle-treated) and IP6-treated BMI1^{High};CHD7^{Low} cells. These data provide compelling evidence that IP6 reverts the increased mTOR activation induced by the BMI1^{High};CHD7^{Low} signature. In line with this finding, we show in Figure 3E that the same concentration of IP6 (1mM) reverts the increased ECAR phenotype found in BMI1^{High};CHD7^{Low} cells.

Fig3 should include the full OCR tracing for BMI1High;CHD7Low G4 MB cells to show in detail the effects of IP6 treatment, as the statement that OCR is not affected is not sufficiently supported by the limited data presented in Fig S3E.

We have now added the full OCR/time plot showing no significant changes in OCR upon 1mM IP6 administration (Figure S3F). This strengthens the conclusion that IP6 acts specifically on the increased ECAR phenotype observed in BMI1^{High};CHD7^{Low} MB cells. Moreover, we have now performed ECAR analysis in control or BMI1^{High};CHD7^{Low} hNSC upon 1mM IP6 administration (see Figure S3I) and we show no significant difference in ECAR/time values, glycolysis and glycolytic capacity, indicating that IP6-mediated inhibition of glycolytic function is specific for the BMI1^{High};CHD7^{Low} tumour context.

ATP studies of Fig 3C and 3G. 3C does not support the interpretation that “BMI1High;CHD7Low G4 MB cells produce significantly more ATP than controls” because ATP content is determined by both ATP production and ATP utilization. What 3C demonstrates is that steady state ATP levels are different in the two groups compared. This same issue of interpretation applies to 3G, where there is also an inconsistency in the conclusion drawn- where in 3C decreased mitochondrial function and increased glycolysis are said to cause increased ATP production, while in 3G increased mitochondrial function and increased glycolysis are said to cause reduced ATP production. The logic that the authors use needs to be more fully explained, or the interpretation needs to be revised. Since differences in ATP utilization have not been considered, and such differences could affect steady state ATP levels, and the interpretation of ATP levels seems problematic, differences in ATP utilization should be incorporated into the interpretation.

We thank the reviewer for this comment which prompted us to investigate ATP utilization to try and achieve a more comprehensive understanding of the observed changes in the ATP level. We have now analysed ADP/ATP ratio, which is widely used to estimate ATP utilization^{15, 16, 17}, in MB and hNSC cells with and without BMI1^{High};CHD7^{Low} signature (Figure 3D and 3I). Interestingly, we found that ADP/ATP ratio is not changed in MB cells irrespective for the presence of the signature; while ADP/ATP ratio is increased in BMI1^{High};CHD7^{Low} hNSC compared to control hNSC.

These new findings, together with the increased level of ATP in steady state, indicate that ATP utilization is not altered in BMI1^{High};CHD7^{Low} MB cells, suggesting that the observed deregulation of OCR and ECAR ultimately impacts on energy production.

On the contrary, the decreased ATP level in steady state found in BMI1^{High};CHD7^{Low} hNSC can be due to an increase ATP utilization, which is not seen in the tumour context.

Professor Silvia Marino

Results presented in lines 225-266 should be broken into at least two paragraphs to enhance readability. These results are now described in two different paragraphs to enhance readability.

Reviewer #2 (expertise: Medulloblastoma models and genetics)

Beautiful paper that correlates BMI expression and CHD7 suppression in a subset of G4 medulloblastoma, and demonstrates resulting mTOR dependent deregulation of inositol metabolism and therapeutic inroads that result. The ability to integrate large datasets and complex data into easily understood figures was impressive. Effects sizes were robust. I had only minor suggestions.

1. For experiments in Fig 2 and going forward, the authors compare G4 lines with their signature to either G4 lines without the signature or with hNSCs. In many cases, it was unclear to me whether the green bars without signature were always hNSCs or may represent signature negative G4 lines. It was also unclear to me which experiments were done in human and which in mouse lines. Can the authors clarify these issues, and can key experiments be repeated in additional signature pos and negative human lines, to assure the generalizability of conclusions?

To enhance clarity we have now used a colour code throughout the manuscript to depict which graphs represent experiments on BMI1^{High};CHD7^{Low} MB cells (red), isogenic MB control cells (green), BMI1^{High};CHD7^{Low} hNSC (purple) and isogenic hNSC control cells (violet) . Mouse cells were not used for any *in vitro* functional assay, murine models were used for the developmental study only.

We certainly agree with the reviewer about the need to carry out the experiments on more than one human MB cell lines and we had indeed used two different MB lines (ICb1299 and CHLA-01-Med) for all the genome-wide analysis (RNA-Seq, DNA methylation and ChIP-Seq), which was then cross-validated with G4 MB tumour samples. Moreover, both lines were used for functional experiments and the mean of the values obtained has been plotted as a single graph representing the two lines for each assay.

2. Also for experiments in Fig 2, the authors conclude that, "These results validate the in silico prediction of an enhanced signalling through the tyrosine-kinase receptors FLT3 and MET in BMI1High;CHD7Low G4 MB but not in hNSC with same molecular signature."

This isn't strictly true, they have shown increased dependence, but can they also show by westerns, that these cells show increased signaling, (and that inhibitors actually block their intended targets) as claimed?

This is a very good point, which we felt had to be addressed experimentally. The receptors tyrosine-kinase (RTK) FLT3 and MET are known activators of the AKT/mTOR signalling^{18, 19, 20} which is overactivated in BMI1^{High};CHD7^{Low} MB cells (Figure 2I), the latter showing increased expression of the two RTKs (Figure S2D). We have now analysed mTOR activation upon treatment with FLT3 and MET inhibitors (Figure S2H,I) demonstrating that they are able to block the pathway specifically in BMI1^{High};CHD7^{Low} MB cells.

3. Can human PDX models tested in in vivo therapy studies in fig 5?

We agree with the reviewer that PDX models derived from BMI1^{High};CHD7^{Low} G4 MB would be the natural next *in vivo* step. Indeed we attempted to perform these experiments and liaised with Marcel Kool, at the DKFZ, who has the largest collection of these models established from several laboratories and checked for the presence of the BMI1^{High};CHD7^{Low} signature among the available G4 MB PDX. We were at first delighted to find two PDX models potentially suitable for our experiments. However we noticed that the BMI1^{High};CHD7^{Low} signature was acquired upon transplantation and tumour formation in mice in one case as it was not present in the primary tumour sample. For the other PDX with signature, the primary tumour sample was not available for analysis (Table 1, A,B

and C, this rebuttal). We decided not to pursue these models, as we reasoned the interpretation of any data would be problematic given the lack of concordance between PDX and primary tumours. We will certainly continue monitoring the scene and aim at carrying out these additional *in vivo* studies as soon as they are feasible, i.e. availability of PDX from BMI1^{High};CHD7^{Low} G4 MB, although this is clearly beyond the scope of this manuscript.

Model	DMB006	CHOPMB-3933	DKFZ-BT251	TK-EP885	TK-MB870	TK-MB913	1387MB
Sample	RCMB6_P3M1	CHOPMB-3933_xp1	DKFZ-BT251	TK-EP885_XP5	TK-MB870_XP4	TK-MB913_p	lcb_1487_XP
Tissue	PDX	PDX	PDX	PDX	PDX	primary	PDX
BMI1 expression	12.11	11.39	10.35	10.91	11.34	10.82	11.13
CHD7 expression	9.74	9.98	9.15	9.69	9.65	8.52	10.3
BMI1 z-score	1.98	0.62	-1.35	-0.30	0.52	-0.46	0.12
CHD7 z-score	0.27	0.77	-0.97	0.17	0.08	-2.31	1.45
BMI1 ^{High} ;CHD7 ^{Low}	no	no	no	no	no	no	no

Model	RCMB45	RCMB45	RCMB52	RCMB35	RCMB38	RCMB38	RCMB49
Sample	RCMB45_P1	RCMB45_pr	RCMB52_xp1	RCMB35_pr	RCMB38_XP	RCMB38_pr	RCMB49_xp1
Tissue	PDX	primary	PDX	primary	PDX	primary	PDX
BMI1 expression	11.61	11.52	11.31	11.63	10.65	10.57	11.62
CHD7 expression	8.95	9.56	9.59	10.11	9.03	9.58	9.36
BMI1 z-score	1.02	0.87	0.45	1.07	-0.79	-0.94	1.05
CHD7 z-score	-1.40	-0.11	-0.06	1.05	-1.23	-0.07	-0.54
BMI1 ^{High} ;CHD7 ^{Low}	yes	no	no	no	no	no	yes

Model	Med610FH	Med-610FH	Med-2312FH	Med-1512FH	Med-2312FH
Sample	Med610FH_pr	Med610FH_P4-2	Med2312FH_XP	Med1512FH_P2-4	Med2312FH_P2-3
Tissue	primary	PDX	PDX	PDX	PDX
BMI1 expression	10.63	10.06	11.03	11.1	10.49
CHD7 expression	10.09	10.42	9.6	9.75	9.57
BMI1 z-score	-0.83	-1.91	-0.07	0.06	-1.09
CHD7 z-score	1.01	1.71	-0.03	0.29	-0.09
BMI1 ^{High} ;CHD7 ^{Low}	no	no	no	no	no

Table 1. Analysis of G4 MB PDX repositories.

BMI1 and CHD7 expression levels obtained from several G4 PDX models and relative primary tumour samples used to evaluate the presence of BMI1^{High};CHD7^{Low} signature.

Reviewer #3, expertise: medulloblastoma models and integrative -omics analysis (Remarks to the Author):

Badodi and colleagues implicate epigenetic regulation of inositol metabolism as a novel mechanism underlying Group 4 MB associated with an expression signature defined by BMI1-high;CHD7-low. A fundamental concern of the study and the majority of the experimental results is the use of in vitro cultured Icb-1299 and CHLA-01-Med cell lines as Group 4 MB cell line models, as these lines appear to be more closely related to Group 3 MB than any other subgroup. Although the authors have previously interpreted these lines as being representative of Group 4 MB (ref. 10), this interpretation is hampered by close inspection of the datasets presented in their prior report (ref. 10) and by more recent studies that have defined these models as Group 3 MB lines. Given this major concern of subgroup identity, it is challenging to appreciate the relevance of the findings presented in the current manuscript, given the uncertainties and doubt concerning biological context. If the author's intent is to implicate a novel mechanism pertinent to Group 4 MB biology, and to nominate potential treatment options for this important subgroup of patients, they are urged to revisit their analyses and validate the primary study conclusions using authentic Group 4 models (i.e. PDX) that have been substantiated as molecularly accurate and representative models of this subgroup.

Professor Silvia Marino

1. By comparing two cultured MB cell lines (Icb1299 and CHLA-01-Med) with or without BMI1-high;CHD7-low signature, the authors examined differentially expressed genes and dysregulated pathways, and identified metabolic adaptation in BMI1-high;CHD7-low MB cell lines which leads to an impaired mitochondrial respiration and enhanced aerobic glycolysis. The authors also showed that IP6 treatment alone or in combination with cisplatin treatment can more specifically inhibit the growth of BMI1-high;CHD7-low MB cell lines compared to control cells. However, there are some fundamental concerns about the use of these cell lines and their inferred identity. First, the subgroup identity of these cell lines is controversial. In one of the previous studies cited here (ref. 10), both Icb-1299 and CHLA-01-Med appear to be more similar to Group 3 MB based on gene expression analysis and were indistinguishable from other Group 3 MB lines. Likewise, in multiple recent studies published by prominent labs (PMID: 33046443; PMID: 31341285), Icb-1299 is annotated as a Group 3 MB. Second, *in vitro* culture of MB cells has been shown to extensively change their gene expression and tumor biology compared with the original tumors or PDXs.

Our group and others^{21, 22, 23, 24} previously used Icb1299 and CHLA-01-Med cells lines as suitable models of the G4 MB subtype; the subgroup affiliation was confirmed by analysing their transcriptome against a classifier²⁵ and via PCA analysis¹. Both approaches confirmed that Icb1299 and CHLA-01-Med belong to G4 with a significant overlap with G3, hence putting them in a boundary of plasticity between G4 and G3, a condition that is now more widely recognised²³ as it was in 2017 but not “more similar to G3 as any other subgroup” as purported by this reviewer. Notably, Hovested and colleagues (PMID: 31341285 cited by the reviewer) define Icb1299 as a G3/G4 model and whilst it is correct that Rusert et al (PMID: 33046443 cited by the reviewer) defines Icb1299 as a G3 PDX model, this conclusion is based solely on tSNE analysis of PDX cells. This is a very different approach to ours, where we used PCA/classifier analysis run on both early passages (cells profiled when isolated from xenografted mice²⁴) and later passages (profiled by us after expansion in culture to obtain enough cells for the functional assays) and show that they all cluster together and retain the molecular subgrouping of the primary tumour sample²⁴.

To pre-empt the valid concern that cultured cells may drift away to some extent from the original tumour, we validated the results obtained in the cell lines on genome-wide datasets obtained from human G4 MB tumour samples. Importantly, we describe many aspects of the biology of these lines which could be validated in datasets of BMI1^{High};CHD7^{Low} G4 MB (e.g. increased proliferation, glycolytic metabolic phenotypes, overactivation of ERK and mTOR pathways, increased aggressivity of the MB formed *in vivo*).

Furthermore, we have now analysed BMI1^{High};CHD7^{Low} G3 MB tumours from published datasets²⁶ (Figure 2A, this rebuttal). In this subgroup the presence of the signature does not affect survival¹ and Pathway analysis of significant differentially expressed genes between tumours with and without the signature do not show an impact on *Superpathway of Inositol Phosphate Compounds* (Figure 2B_[SB2], this rebuttal), hence confirming that the mechanisms described in our study are specifically pertinent to the G4 subgroup.

Figure 2. Pathway analysis of BMI1^{High};CHD7^{Low} G3 MB .

A. Pie chart representing percentage of G3 MB tumour samples with (yellow) or without (gray) the BMI1^{High};CHD7^{Low} signature. B. Histograms showing -log₁₀(FDR) values of canonical pathways differentially enriched between G3 MB patients with or without BMI1^{High};CHD7^{Low} signature identified in RNA-Seq analysis. Canonical pathways are classified based on IPA categories list and colour coded accordingly.

Validation of the major conclusions presented in the study using more physiologically relevant models (i.e. bona fide Group 4 MB PDX) is important and required to effectively substantiate the findings reported here.

We agree with the reviewer that PDX models derived from BMI1^{High};CHD7^{Low} G4 MB would be the natural next *in vivo* step. Indeed we attempted to perform these experiments and liaised with Marcel Kool, at the DKFZ, who has the largest collection of these models established from several laboratories and checked for the presence of the BMI1^{High};CHD7^{Low} signature among the available G4 MB PDX. We were at first delighted to find two PDX models potentially suitable for our experiments. However we noticed that the BMI1^{High};CHD7^{Low} signature was acquired upon transplantation and tumour formation in mice in one case as it was not present in the primary tumour sample. For the other PDX with signature, the primary tumour sample was not available for analysis (Table 1, A,B and C, this rebuttal). We decided not to pursue these models, as we reasoned the interpretation of any data would be problematic given the lack of concordance between PDX and primary tumours. We will certainly continue monitoring the scene and aim at carrying out these additional *in vivo* studies as soon as they are feasible, i.e. availability of PDX from BMI1^{High};CHD7^{Low} G4 MB but these experiments are at this moment in time not possible. We would like to reiterate that we validated the results obtained in the cell lines on genome-wide datasets obtained from human G4 MB tumour samples, hence providing robust evidence of their physiological relevance.

2. In Figure 1A-C, the authors implicate an increased proportion of Eomes-positive cells concomitant with a reduction in NeuN-positive cells in the developing cerebella of Math1Cre;STOPFloxBmi1;Chd7f/+ transgenic mice. Although the authors quantify these differences compared to relevant controls, the proportional differences in these populations appear to be marginal at best based on representative IHC. The authors interpret these differences as a potential lineage shift towards an expansion of the UBC lineage and reduction in the granule neuron lineage. However, these inferences are based on the IHC-derived expression patterns of two lineage markers (Eomes, NeuN). More comprehensive analyses are required to substantiate these claims. The authors should perform single-cell RNA sequencing analysis of the developing cerebella in both the control and Math1Cre;STOPFloxBmi1;Chd7f/+ transgenics to support their current findings.

We did not observe MB in the compound mutant mice generated to assess the potential to initiate tumourigenesis by the BMI1/CHD7 molecular convergence in Math1+ progenitors. It is beyond the scope of this paper, which focuses on MB, to comprehensively characterise the impact of these mutations on cerebellar development. However, because recent work has shown that UBC, which are derived from Math1+ progenitors as well as GC, are the putative cell-of-origin of G4 MB^{4, 23, 27}, we set out to assess the specific impact of the genetic manipulation on these two defined cell types. We have taken a scholar approach, which relies on analysing and quantifying these cells on the basis of assessing the expression of well-established markers, which are specific for these population in the developing cerebellum. This approach allows us to show hyperplasia of UBC progenitors which is a strong indicator of the role these two oncogenic events play in the pathogenesis of G4 MB subgroup. It also prompted us to explore other BMI1^{High};CHD7^{Low}-mediated MB-specific molecular deregulations which must have been essential for tumour growth and we discovered the role of IP6.

Hence, although scRNA-Seq analysis, or even better spatial transcriptomic, would be a very interesting approach to comprehensively analyse the cerebellar phenotype of these mice, this is beyond the scope of the present study.

3. In Figure 1G and 4A, the authors indicate that CHD7 knockdown or IP6 treatment reduced cell proliferation by examining the number of viable cells during culture. However, this method cannot distinguish proliferation vs apoptosis. The authors should perform BrdU incorporation or apoptosis assays (i.e. annexin V staining) to determine whether cell proliferation or apoptosis was affected under these experimental conditions.

We have now analysed apoptosis of MB cells with or without BMI1^{High};CHD7^{Low} signature treated with increasing concentration of IP6 and we did not find an increase of apoptotic cells (Figure S4B). This is in line with the *in vivo* observation that IP6 treatment did not impact the number of cleaved Casp-3 positive cells (Figure 5I). On the

Professor Silvia Marino

contrary, the decreased percentage of Ki-67 positive cells *in vivo* (Figure 5H) and the proliferation assay *in vitro* (Figure 4A), suggest that IP6 is mediating a cytostatic effect, as previously described also in other tumour contexts^{11, 12}.

4. Aerobic glycolysis has been shown to be a therapeutic target in Group 3 MB. Genes involved in aerobic glycolysis are also highly expressed in Group 3 MB compared with other subgroups (PMID: 30862721). In the current study, the authors showed enhanced aerobic glycolysis in BMI1-high;CHD7-low MB cells compared with control cells. To further confirm these findings in human patient samples, the authors should examine the expression of glycolytic genes in Group 4 MB with or without the BMI1-high;CHD7-low signature.

This is a very good point and we thank the reviewer for this suggestion. We have now analysed the expression of glycolytic genes in G4 MB tumour samples. Interestingly, we found increased expression of *HK2*, *PFKP*, *ENO4*, *PDK1* and *LDHB*, encoding key enzymes regulating glucose metabolism, in BMI1^{High};CHD7^{Low} G4 MB tumour samples as compared to G4 MB without the signature (Figure S3E). This is in line with the increased level of valine and leucine detected by MRS in BMI1^{High};CHD7^{Low} G4 MB tumour samples (Figure S3D) and strengthens the cross-validation of the glycolytic phenotype between our *in vitro* model and G4 MB tumours with the same signature. Importantly, the expression of glycolytic genes is not significantly changed between G3 MB patients with or without signature (Figure 3, this rebuttal) confirming the specific role mediated by BMI1^{High};CHD7^{Low} in the G4 MB subgroup.

References

1. Badodi S, *et al.* Convergence of BMI1 and CHD7 on ERK Signaling in Medulloblastoma. *Cell Rep* **21**, 2772-2784 (2017).
2. Moreno-Sanchez R, Rodriguez-Enriquez S, Saavedra E, Marin-Hernandez A, Gallardo-Perez JC. The bioenergetics of cancer: is glycolysis the main ATP supplier in all tumor cells? *Biofactors* **35**, 209-225 (2009).

3. Weyer A, Schilling K. Developmental and cell type-specific expression of the neuronal marker NeuN in the murine cerebellum. *J Neurosci Res* **73**, 400-409 (2003).
4. Vladoiu MC, *et al.* Childhood cerebellar tumours mirror conserved fetal transcriptional programs. *Nature* **572**, 67-73 (2019).
5. Schroterova L, Jezkova A, Rudolf E, Caltova K, Kralova V, Hanusova V. Inositol hexaphosphate limits the migration and the invasiveness of colorectal carcinoma cells in vitro. *Int J Oncol* **53**, 1625-1632 (2018).
6. Yang GY, Shamsuddin AM. IP6-induced growth inhibition and differentiation of HT-29 human colon cancer cells: involvement of intracellular inositol phosphates. *Anticancer Res* **15**, 2479-2487 (1995).
7. Shamsuddin AM, Yang GY, Vucenik I. Novel anti-cancer functions of IP6: growth inhibition and differentiation of human mammary cancer cell lines in vitro. *Anticancer Res* **16**, 3287-3292 (1996).
8. Deliliers GL, *et al.* Effect of inositol hexaphosphate (IP(6)) on human normal and leukaemic haematopoietic cells. *Br J Haematol* **117**, 577-587 (2002).
9. Agarwal C, Dhanalakshmi S, Singh RP, Agarwal R. Inositol hexaphosphate inhibits growth and induces G1 arrest and apoptotic death of androgen-dependent human prostate carcinoma LNCaP cells. *Neoplasia* **6**, 646-659 (2004).
10. Ren L, *et al.* Metabolomics uncovers a link between inositol metabolism and osteosarcoma metastasis. *Oncotarget* **8**, 38541-38553 (2017).
11. Bizzarri M, Dinicola S, Bevilacqua A, Cucina A. Broad Spectrum Anticancer Activity of Myo-Inositol and Inositol Hexakisphosphate. *Int J Endocrinol* **2016**, 5616807 (2016).
12. Vucenik I, Shamsuddin AM. Cancer inhibition by inositol hexaphosphate (IP6) and inositol: from laboratory to clinic. *J Nutr* **133**, 3778S-3784S (2003).
13. Grases F, Simonet BM, Prieto RM, March JG. Phytate levels in diverse rat tissues: influence of dietary phytate. *Br J Nutr* **86**, 225-231 (2001).
14. Agte V, Jahagirdar M, Chiplonkar S. Apparent absorption of eight micronutrients and phytic acid from vegetarian meals in ileostomized human volunteers. *Nutrition* **21**, 678-685 (2005).
15. Zanutelli MR, *et al.* Regulation of ATP utilization during metastatic cell migration by collagen architecture. *Mol Biol Cell* **29**, 1-9 (2018).
16. Jeneson JA, Westerhoff HV, Kushmerick MJ. A metabolic control analysis of kinetic controls in ATP free energy metabolism in contracting skeletal muscle. *Am J Physiol Cell Physiol* **279**, C813-832 (2000).
17. Bressan C, *et al.* The dynamic interplay between ATP/ADP levels and autophagy sustain neuronal migration in vivo. *Elife* **9**, (2020).

Professor Silvia Marino

18. Chen W, *et al.* mTOR signaling is activated by FLT3 kinase and promotes survival of FLT3-mutated acute myeloid leukemia cells. *Mol Cancer* **9**, 292 (2010).
19. Demkova L, Kucerova L. Role of the HGF/c-MET tyrosine kinase inhibitors in metastatic melanoma. *Mol Cancer* **17**, 26 (2018).
20. Lam BQ, Dai L, Qin Z. The role of HGF/c-MET signaling pathway in lymphoma. *J Hematol Oncol* **9**, 135 (2016).
21. Frisira E, *et al.* NPI-0052 and gamma-radiation induce a synergistic apoptotic effect in medulloblastoma. *Cell Death Dis* **10**, 785 (2019).
22. Manoranjan B, *et al.* Wnt activation as a therapeutic strategy in medulloblastoma. *Nat Commun* **11**, 4323 (2020).
23. Hovestadt V, *et al.* Resolving medulloblastoma cellular architecture by single-cell genomics. *Nature* **572**, 74-79 (2019).
24. Zhao X, *et al.* Global gene expression profiling confirms the molecular fidelity of primary tumor-based orthotopic xenograft mouse models of medulloblastoma. *Neuro Oncol* **14**, 574-583 (2012).
25. Robinson G, *et al.* Novel mutations target distinct subgroups of medulloblastoma. *Nature* **488**, 43-48 (2012).
26. Cavalli FMG, *et al.* Intertumoral Heterogeneity within Medulloblastoma Subgroups. *Cancer Cell* **31**, 737-754 e736 (2017).
27. Lin CY, *et al.* Active medulloblastoma enhancers reveal subgroup-specific cellular origins. *Nature* **530**, 57-62 (2016).

Reviewers' Comments:

Reviewer #1:

Remarks to the Author:

The authors have thoughtfully considered each of the points in my initial review, made appropriate changes and clarifications, and provided cogent responses in their rebuttal letter. Additional experimental studies have strengthened the work. The cohesion between Figure 1 and the rest of the paper could still be improved by making the logical flow more clear in the text. This suggested change requires only minor editing and the manuscript is otherwise excellent and appropriate for publication.

Reviewer #2:

Remarks to the Author:

revised manuscript adequately addresses issues raised in review.

Reviewer #3:

Remarks to the Author:

In the revised manuscript, the authors provided a few new pieces of evidence to address some of the reviewers' suggestions. However, two fundamental concerns about the misinterpretation of the two Group 3 MB cell lines as Group 4 cell lines and the lack of validation using authentic Group 4 models were left unresolved. Thus, I could not further appreciate the relevance of the findings presented in the revised manuscript.

1. A major concern of this study remains the misinterpretation of Icb1299 and CHLA-01-Med as Group 4 MB cell lines. In their rebuttal, the authors erroneously state that Hovestadt and colleagues defined Icb1299 as a G3/G4 model (PMID: 31341285). However, this is not the case. In that study, by DNA methylation classification – the current gold-standard for assigning MB subgroup status, Icb1299 was clearly characterized as a Group 3, subtype II tumor (PMID: 31341285; Supplementary Table 1). In the original paper that described the Icb1299 model (PMID: 22459127), the model is described as either 'Group C' or 'Group D', depending on the passage, based on gene expression analysis; these are previous generation annotations for Group 3 and Group 4, respectively, and expression profiling-based classification methods have been superseded by more robust DNA methylation arrays (PMID: 29539639). In the more recent papers that have been cited by the authors, the Icb1299 line is not experimentally characterized as a Group 3 or Group 4 model, rather, subgroup status has merely been assumed as either Group 3/Group 4 (PMID: 22459127) or Group 4 (PMID: 22459127).

Although Group 3 and Group 4 MB are known to exhibit some degree of molecular and cellular overlap, the majority of Group 3 and Group 4 MB tumors can be readily discriminated and exhibit highly distinctive biology (PMID: 31341285; PMID: 28726821; PMID: 28609654). Considering Group 3 and Group 4 MB are broadly molecularly and biologically distinct diseases, understanding their biology using authentic models is critical. To this end, the misinterpretation of Group 3 or Group 4 cell models can be unfortunately misleading to the community and, continued publication as such, runs the risk of continuing to propagate erroneous information in future studies.

2. As major conclusions of the manuscript are based on the study of in vitro cultured Icb1299 and CHLA-01-Med cells which are most likely Group 3 MB cell lines as discussed above, validation of the key findings using authentic Group 4 PDX models is both important and required to appreciate the main study findings. Although this concern has been raised by multiple reviewers, no further evidence was provided. A large number of Group 4 MB PDX lines have been established in the community (PMID: 30349086; PMID: 31693904; PMID: 33046443; PMID: 32519082). In the

rebuttal, the authors state that the current Group 4 PDX lines either do not express the signature or the signature is not consistent between the primary tumor and PDX. However, it is feasible to manipulate the PDXs shortly in vitro (e.g., by overnight virus infection to overexpress BMI1 and/or silence CHD7; analogous to what has been done in the Icb1299 and CHLA-01-Med MB cell lines used in the current manuscript) to mimic the signature and then compare them with their isogenic control for a more physiologically relevant in vivo functional validation. Such studies would add important credence to this work, that remains otherwise challenging to appreciate given the persistent confusion regarding the subgroup status of the Icb1299 and CHLA-01-Med cell lines.

Reviewer#1 report

The authors have thoughtfully considered each of the points in my initial review, made appropriate changes and clarifications, and provided cogent responses in their rebuttal letter. Additional experimental studies have strengthened the work. The cohesion between Figure 1 and the rest of the paper could still be improved by making the logical flow more clear in the text. This suggested change requires only minor editing and the manuscript is otherwise excellent and appropriate for publication.

We thank the reviewer for appreciating our revised version of the manuscript and the additional experimental studies performed. We have now further clarified in the text (page 5-6) the logical flow that prompt us to explore the role of BMI1 and CHD7 through a genome-wide analysis in cell models, integrated with publicly available G4 MB datasets, starting from the developmental study presented in Figure 1.

Reviewer#2 report

Revised manuscript adequately addresses issues raised in review.

We thank the reviewer for appreciating our revision.

Reviewer#3 report

In the revised manuscript, the authors provided a few new pieces of evidence to address some of the reviewers' suggestions. However, two fundamental concerns about the misinterpretation of the two Group 3 MB cell lines as Group 4 cell lines and the lack of validation using authentic Group 4 models were left unresolved. Thus, I could not further appreciate the relevance of the findings presented in the revised manuscript.

This reviewer fails to acknowledge that we have validated all results obtained in the cell models on the G4 MB tumour samples. This point is a major strength of our study. It is disappointing that this reviewer continues to ignore this fundamental result.

1. A major concern of this study remains the misinterpretation of Icb1299 and CHLA-01-Med as Group 4 MB cell lines. In their rebuttal, the authors erroneously state that Hovestadt and colleagues defined Icb1299 as a G3/G4 model (PMID: 31341285). However, this is not the case. In that study, by DNA methylation classification – the current gold-standard for assigning MB subgroup status, Icb1299 was clearly characterized as a Group 3, subtype II tumor (PMID: 31341285; Supplementary Table 1). In the original paper that described the Icb1299 model (PMID: 22459127), the model is described as either 'Group C' or 'Group D', depending on the passage, based on gene expression analysis; these are previous generation annotations for Group 3 and Group 4, respectively, and expression profiling-based classification methods have been superseded by more robust DNA methylation arrays (PMID: 29539639). In the more recent papers that have been cited by the authors, the Icb1299 line is not experimentally characterized as a Group 3 or Group 4 model, rather, subgroup status has merely been assumed as either Group 3/Group 4 (PMID: 22459127) or Group 4 (PMID: 22459127).

Although Group 3 and Group 4 MB are known to exhibit some degree of molecular and cellular overlap, the majority of Group 3 and Group 4 MB tumors can be readily discriminated and exhibit highly distinctive biology

Professor Silvia Marino

(PMID: 31341285; PMID: 28726821; PMID: 28609654). Considering Group 3 and Group 4 MB are broadly molecularly and biologically distinct diseases, understanding their biology using authentic models is critical. To this end, the misinterpretation of Group 3 or Group 4 cell models can be unfortunately misleading to the community and, continued publication as such, runs the risk of continuing to propagate erroneous information in future studies.

2. As major conclusions of the manuscript are based on the study of *in vitro* cultured Icb1299 and CHLA-01-Med cells which are most likely Group 3 MB cell lines as discussed above, validation of the key findings using authentic Group 4 PDX models is both important and required to appreciate the main study findings. Although this concern has been raised by multiple reviewers, no further evidence was provided. A large number of Group 4 MB PDX lines have been established in the community (PMID: 30349086; PMID: 31693904; PMID: 33046443; PMID: 32519082). In the rebuttal, the authors state that the current Group 4 PDX lines either do not express the signature or the signature is not consistent between the primary tumor and PDX. However, it is feasible to manipulate the PDXs shortly *in vitro* (e.g., by overnight virus infection to overexpress BMI1 and/or silence CHD7; analogous to what has been done in the Icb1299 and CHLA-01-Med MB cell lines used in the current manuscript) to mimic the signature and then compare them with their isogenic control for a more physiologically relevant *in vivo* functional validation. Such studies would add important credence to this work, that remains otherwise challenging to appreciate given the persistent confusion regarding the subgroup status of the Icb1299 and CHLA-01-Med cell lines.

We agree that there is conflicting literature on the subgroup allocation of these patient-derived MB lines, this is largely due to the recently recognised plasticity of the G3/G4 subgroup allocation, which recapitulates the MB tumour subgroup classification^{1,2}. We acknowledge this limitation in the MS (now even more extensively, see page 6) and we have removed any reference to the subgroup when describing experiments with the cell model. Models have strengths and limitations and it may well be that other features of G4 MB are not recapitulated by these cell lines. It is also possible that for other aspects these cell lines reflect biological characteristics of G3 MB; this is indeed what would be predicted on the basis of the well recognised plasticity of the G3/G4 MB subgroup. However, this is the best model currently available of the BMI1^{High};CHD7^{Low} signature and we have validated every single conclusion of this paper on G4 MB tumour samples. We are now referring to G4 MB subgroup only when describing data directly obtained from the tumour samples.

To ensure we were as unbiased as possible in our analysis, we have also analysed G3 MB tumour samples with and without the BMI1^{High};CHD7^{Low} signature to assess whether the results we had obtained in the cell models would apply to G3 MB. We provide compelling evidence that this is not the case, there is no impact on patients' survival or on the *Superpathway of Inositol Phosphate Compounds*, and no significant changes in the expression of glycolytic genes in BMI1^{High};CHD7^{Low} G3 MB tumours. The latter experiment was suggested by this reviewer and has provided additional support to our interpretation. We agree with the editorial suggestion to add these data to the MS (Figure S2C and Figure S3F and page 6 and 8) to ensure this analysis is available to the readers immediately and not only upon consultation of the point-by-point response to reviewers, which I understand will be public in line with Nature Communications editorial policy.

In summary, we provide compelling evidence that modelling the BMI1^{High};CHD7^{Low} signature in Icb1299 and CHLA-01-Med cells reflect biological features of the G4 MB subgroup, as assessed on G4 MB tumour samples. We have been commended about this fundamental strength of our manuscript by both reviewer 1 and 2.

References

1. Hovestadt V, *et al.* Resolving medulloblastoma cellular architecture by single-cell genomics. *Nature* **572**, 74-79 (2019).
2. Kumar R, *et al.* Clinical Outcomes and Patient-Matched Molecular Composition of Relapsed Medulloblastoma. *J Clin Oncol*, JCO2001359 (2021).

Reviewers' Comments:

Reviewer #4:

Remarks to the Author:

The authors have adequately accommodated my suggestions related to the classification of the medulloblastoma cell line and PDX models in their revised manuscript.